# Reasoning-preserved Efficient Distillation of Large Language Models via Activation-aware Initialization

**Junlin He** [@][1]   **Yihong Tang** [@][2]   **Tong Nie** [1]   **Guilong Li** [1]   **Binyu Yang** [1]   **Jinxiao Du** [1]   **Lijun Sun** [2]   **Wei Ma** [✉][1]

## Abstract

Efficient Distillation (EDistill) compresses large language models (LLMs) by structured pruning parameters and tuning lightweight modules with high training efficiency. Although these EDistilled LLMs achieve state-of-the-art (SOTA) performance on general ability benchmarks relative to similarly sized LLMs, we identify a severe degradation in their multi-step reasoning ability, which we term *reasoning collapse*. We systematically analyze the geometric origins of *reasoning collapse* and show that the SOTA EDistill method based on width-reducing projection matrices suffers from *eRank collapse*, in which the effective rank (eRank) of hidden representations drops. We theoretically explain how singular values of randomly initialized projection matrices become unevenly distributed, leading to *eRank collapse* and thus token indistinguishability. To address this issue, we propose **RED** (**R**easoning-preserved **E**fficient **D**istillation) for LLMs, which introduces *activation-aware initialization* to initialize projection matrices as channel-selection matrices, thus theoretically mitigating *eRank collapse*. Experiments on Llama and Qwen series demonstrate that RED substantially recovers reasoning while maintaining high training efficiency and SOTA general ability.

## 1. Introduction

Large Language Models (LLMs) have achieved impressive performance across a wide spectrum of Natural Language Processing (NLP) tasks (Achiam et al., 2023; Dubey et al., 2024; Guo et al., 2025; Team et al., 2024; Yang et al., 2025). However, their deployment in real-world applications is of-

[1]The Hong Kong Polytechnic University, Hong Kong SAR, China [2]McGill University, Montreal, QC, Canada. [@]Equal Contribution. Correspondence to: Wei Ma <wei.w.ma@polyu.edu.hk>.

*Proceedings of the 43rd International Conference on Machine Learning*, Seoul, South Korea. PMLR 306, 2026. Copyright 2026 by the author(s).

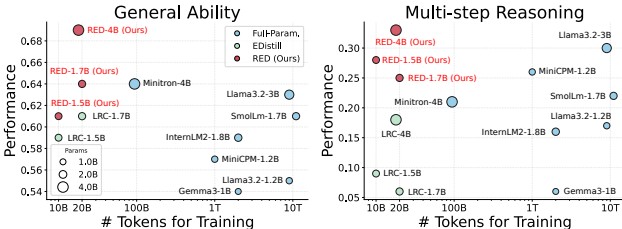

*Figure 1.* Training efficiency vs. performance. General ability (left) and multi-step reasoning (right) performance versus training tokens for Full-parameter training SLMs (Base versions), EDistill baselines, and our method, RED. While existing EDistill methods excel on general tasks, they exhibit a sharp performance drop on reasoning tasks, which our method successfully mitigates.

ten constrained by prohibitive memory and computational costs (Thompson et al., 2020). This has catalyzed the development of Small Language Models (SLMs) that aim to retain strong ability while operating within significantly reduced resource budgets.

Despite their compact size, training high-performing SLMs from scratch remains an extremely resource-intensive endeavor. For instance, training SmolLM2-1.7B requires 11 trillion tokens and takes several weeks on hundreds of A100 GPUs, costing approximately $250,000 (Allal et al., 2025). To mitigate these costs, structured pruning followed by post-pruning recovery has emerged as a promising paradigm (Zhu et al., 2024; Wang et al., 2024; van der Ouderaa et al., 2023; Hu et al., 2024b; Ramesh et al., 2023). It typically involves compressing a large teacher model into a hardware-friendly student model and subsequently retraining it to recover performance. Within this paradigm, recovery strategies can be broadly categorized into: full-parameter retraining (Xia et al., 2023; Muralidharan et al., 2024; Dubey et al., 2024), which demands substantial computational budgets; LoRA-style retraining (Ma et al., 2023; Ashkboos et al., 2024), which offers parameter efficiency but often fails to close the fidelity gap with the teacher (Chen et al., 2024); and efficient distillation (EDistill) via lightweight modules.

Such EDistill methods either merge multiple Transformer layers into one learnable layer to reduce depth (*depth-reduced*) (Chen et al., 2024; Shopkhoev et al., 2025), or insert learnable projection matrices (Hao et al., 2025) into the teacher model to reduce the hidden size (*width-reduced*). During training of these methods, all parameters of teacher

models are frozen, achieving state-of-the-art (SOTA) results on general ability benchmarks relative to similarly sized LLMs with minimal data and computational resources.

However, we observe that EDistilled LLMs suffer from a severe performance degradation in multi-step reasoning tasks, such as *Mathematical Reasoning* and *Code Generation* (see Figure 1, right). We term this phenomenon *reasoning collapse* (Section 4). Moreover, we show that the SOTA *width-reduced* EDistill method (i.e., LRC), using projection matrices, suffers from *eRank collapse* (a sharp decline in *effective rank* in hidden representations). Through theoretical analysis, we show that when employing learnable projection matrices for dimension alignment, the gradients of the projection matrices' singular values are proportional to those singular values. Consequently, certain singular values exhibit exponential growth, leading to the space of projection matrices and representations being dominated by a few large directions. Therefore, tokens become increasingly indistinguishable, which fundamentally undermines multi-step reasoning (Fu et al., 2026).

To address this issue, we propose **R**easoning-preserved **E**fficient **D**istillation (**RED**). RED introduces **activation-aware initialization** for projection matrices, initializing projection matrices as channel-selection matrices. At initialization, each singular value of projection matrices is set to 1, and gradients become 0, thereby avoiding the uneven singular value distribution of projection matrices caused by random initialization and thus mitigating *eRank collapse*. To select important channels, we collect activations from the teacher model on a small calibration dataset and estimate the importance of each channel based on these activations. This approach enables the student model to preserve the most critical subspaces during initialization, rather than obscuring them with random matrices. Under identical training conditions as LRC, RED achieves SOTA performance in both general ability and multi-step reasoning benchmarks compared to full-parameter, LoRA-style retraining and EDistill methods. Our contributions are as follows:

- We are the first to identify and systematically investigate *reasoning collapse* in EDistilled LLMs through the lens of representation geometry.

- We provide a theoretical explanation for how randomly initialized projection matrices of the *width-reduced* EDistill method lead to *eRank collapse* and thus token indistinguishability, undermining the multi-step reasoning ability.

- We propose RED, which utilizes activation-aware initialization to theoretically mitigate the eRank collapse of hidden representations. We empirically demonstrate that projection matrices of RED show an even distribution of singular values and RED recovers reasoning performance while maintaining high efficiency and superior general ability on both the Llama and Qwen series.

## 2. Related Work

**Pruning and Recovery.** Structured pruning aims to derive hardware-efficient student models by removing entire groups of parameters from teacher LLMs (Zhu et al., 2024; Hu et al., 2024b). To mitigate the resulting performance loss, a recovery stage is introduced. Full-parameter retraining methods (Sheared-LLaMA (Xia et al., 2023), Minitron (Muralidharan et al., 2024), Llama 3.2 (Dubey et al., 2024)) recover ability via retraining on trillions of tokens. The computational overhead is often prohibitive. Conversely, LoRA-style retraining methods (Ma et al., 2023; Ashkboos et al., 2024) offer lower costs but struggle to recover the teacher's performance (Chen et al., 2024).

**EDistill via Lightweight Modules.** A more recent paradigm focuses on EDistill by training only a fraction of newly inserted lightweight modules while freezing other inherited parameters. This includes *depth-reduced* EDistill methods (e.g., LLMStreamline (Chen et al., 2024), ReplaceMe (Shopkhoev et al., 2025)) that utilize a learnable single layer to replace multiple layers, and *width-reduced* EDistill methods (e.g., LRC (Hao et al., 2025)) that employ projection matrices for dimension alignment. While these methods achieve SOTA results on general ability benchmarks with minimal training resources, we uncover a critical flaw: *reasoning collapse* of EDistilled LLMs.

**Reasoning Collapse in LLMs.** *Reasoning collapse* has been recognized in scenarios such as model quantization (Li et al., 2025; Liu et al., 2025) and training-free structured pruning of LLMs (Lele et al.). However, the phenomenon of reasoning collapse in EDistilled LLMs remains underexplored, with prior work focusing solely on their general ability. Moreover, we utilize the geometry of representations (Park et al., 2023; Wei et al., 2024; Skean et al., 2025) to investigate why reasoning collapse occurs in EDistilled LLMs. We further conduct a theoretical analysis of the behavior of width-reduced EDistilled LLMs and propose an activation-aware initialization strategy.

## 3. Preliminaries

In this section, we introduce the architectural formulation of LLMs and establish the notation used throughout the paper.

### 3.1. Architecture of LLMs

Given an input token sequence of length $L$, an LLM first maps discrete tokens into continuous representations using an embedding matrix $W_e \in \mathbb{R}^{\text{Voc} \times D}$, resulting in the initial hidden representations $X_0 \in \mathbb{R}^{L \times D}$, where Voc is the vocabulary size and $D$ is the hidden dimension. The representations are then processed by a stack of $N$ Transformer blocks. Each layer follows a Pre-Norm architecture and consists of a multi-head self-attention (MHSA) and a feed-forward network (FFN) modules, both equipped with residual connections (Dubey et al., 2024). Let $X_{i-1}$ and $X_i$ denote the input and output of the $i$-th Transformer layer

(Trans.). The forward computation is given by:

$$H_i = X_{i-1} + \text{MHSA}_i(\text{Norm}_{a,i}(X_{i-1})),$$
$$X_i = H_i + \text{FFN}_i(\text{Norm}_{f,i}(H_i)), \qquad (1)$$

where $\text{Norm}_{a,i}$ and $\text{Norm}_{f,i}$ denote the normalization layers applied before the $\text{MHSA}_i$ and $\text{FFN}_i$ modules respectively. In modern LLMs, these normalization layers are typically implemented using RMSNorm (Zhang & Sennrich, 2019) (see Appendix I.1). After $N$ Transformer blocks, the final hidden representations $X_N \in \mathbb{R}^{L \times D}$ are normalized and projected to vocabulary logits $Z_N \in \mathbb{R}^{L \times \text{Voc}}$ through the unembedding layer $W_u \in \mathbb{R}^{D \times \text{Voc}}$:

$$Z_N = \text{Norm}_{\text{final}}(X_N) W_u. \qquad (2)$$

Applying the $\text{softmax}$ function to $Z_N$ yields the next-token probability distribution over the vocabulary.

### 3.2. EDistill of LLMs

EDistill of LLMs restricts the optimization to the newly inserted lightweight modules while reducing the depth or width of the teacher's architecture. To optimize these lightweight modules, EDistill typically employs a combination of training objectives (see Appendix B).

In *depth-reduced* EDistill, the student contains fewer Transformer blocks. Consecutive teacher blocks $\{i, \dots, i+R-1\}$ are identified as redundant using a small calibration dataset $\mathcal{D}_{\text{pre}}$ and a predefined target number of layers $R$ to remove. The optimal start index $i^\star \in \{1, \dots, N-R+1\}$ is selected based on minimal degradation in perplexity or maximal similarity between the input $X_{i^\star}$ and output $X_{i^\star+R-1}$ representations. Efficiency is typically achieved by replacing them with a single learnable Transformer layer (Chen et al., 2024) or linear layer (Shopkhoev et al., 2025).

In *width-reduced* EDistill, the student model operates on a reduced residual dimension $D' < D$ while preserving the internal structures of MHSA and FFN modules. This is achieved by introducing learnable projection matrices that map representations between the reduced and original dimensions while keeping all parameters of the teacher model frozen: for the $i$-th Transformer layer, up-projection matrices $O_i^a, O_i^f \in \mathbb{R}^{D' \times D}$ are applied after normalization, and down-projection matrices $Q_i^a, Q_i^f \in \mathbb{R}^{D \times D'}$ are applied after $\text{MHSA}_i$ and $\text{FFN}_i$ modules. Let $X'_{i-1}$ and $X'_i$ denote the input and output of the student model's $i$-th Transformer layer. The forward computation is given by

$$H'_i = X'_{i-1} + \text{MHSA}_i\left(\text{Norm}'_{a,i}(X'_{i-1})O_i^a\right) Q_i^a,$$
$$X'_i = H'_i + \text{FFN}_i\left(\text{Norm}'_{f,i}(H'_i)O_i^f\right) Q_i^f, \qquad (3)$$

where $\text{Norm}'_{a,i}$ and $\text{Norm}'_{f,i}$ are the corresponding re-initialized normalization layers of student models, and the embedding and unembedding layers are similarly adapted using projection matrices to ensure compatibility with the teacher's vocabulary space. After training these learnable projection matrices, they are merged with the frozen parameters to obtain the final model.

## 4. Reasoning Collapse in EDistilled LLMs

EDistilled LLMs suffer from severe performance degradation in multi-step reasoning tasks, a phenomenon we term *reasoning collapse*. This degradation persists even under standard retraining, severely hindering the application of distilled LLMs in broader scenarios.

*Table 1.* Comparison of depth reduction on GSM8K. "Layers (Red.)" denotes the remaining layers and the reduction ratio.

| Method | Teacher (layers) | Module | Layers (Red.) | GSM8K |
|---|---|---|---|---|
| ReplaceMe | Qwen2.5-7B-Ins. (28 layers) | Linear | 26 (7.1%) | 0.59 |
| | | | 24 (14.3%) | **0.29** |
| | | | 22 (21.4%) | **0.04** |
| | Llama3.1-8B-Ins. (32 layers) | | 28 (12.5%) | 0.63 |
| | | | 26 (18.8%) | **0.49** |
| | | | 24 (25.0%) | **0.14** |
| LLM Streamline | Llama3.1-8B-Base (32 layers) | Trans. | 26 (18.8%) | 0.41 |
| | | | 20 (37.5%) | **0.24** |
| | | | 18 (43.8%) | **0.09** |

**Reasoning Collapse in Depth-reduced EDistilled LLMs.** As shown in Table 1, regardless of the training method, replacement modules, or teacher model employed, once the depth is reduced beyond a certain threshold, multi-step reasoning ability drops precipitously. This is primarily attributed to the depth dependency of multi-step reasoning, which limits the number of layers that LLMs can discard (Liu et al., 2022; Gupta et al., 2025; Tan et al., 2025; Heakl et al., 2025). By further analyzing the co-evolution of maximum token representation similarity and predictive uncertainty, we demonstrate that the student model is forced to compress the entire transition from exploration to commitment into a narrow computational window, and the student's representations become markedly less separable precisely at the point of decision-making (Appendix E).

**Reasoning Collapse in Width-reduced EDistilled LLMs.** Similar *reasoning collapse* also occurs in width-reduced EDistilled LLMs. Unlike depth reduction, which compresses reasoning in *time*, width reduction constrains computation in *space* by forcing hidden representations to lie in a lower-dimensional subspace. To further examine the geometry of compressed representations, given the hidden representations $X \in \mathbb{R}^{L \times D}$, following Wei et al. (2024), the representation matrix is zero-centered ($\sum_{l=1}^{L} X^l = \mathbf{0}$) and row-normalized ($\|X^l\|_2 = 1$), where $X^l$ is the feature of the $l$-th token. Let $\Sigma = \frac{1}{L} X^\top X$ denote the covariance matrix with eigenvalues $\{\lambda_j\}_{j=1}^{D}$. We define eRank as:

$$\text{eRank}(X) = \exp\left(-\sum_{j=1}^{D} p_j \log p_j\right), \qquad (4)$$

where $p_j = \frac{\lambda_j}{\sum_{k=1}^{D} \lambda_k}$. A smaller eRank indicates that representations are dominated by a few principal directions, reflecting reduced expressivity (He et al., 2024).

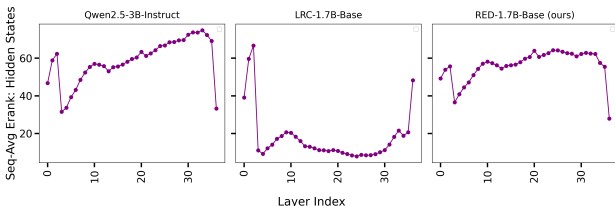

*Figure 2.* Comparison of eRank across layers for the teacher model (left), the LRC (middle), and our RED (right). Sample sequences are from Nemotron-Pretraining-Dataset.

As shown in Figure 2, compared to the teacher model (Qwen2.5-3B-Instruct, left), the width-reduced student LRC-1.7B-Base (middle), which represents a SOTA approach to width-reduced EDistill, exhibits a substantially smaller eRank across most layers. This reduction in intrinsic dimensionality coincides with a sharp degradation in reasoning performance, with GSM8K accuracy dropping to **0.09** (see Table 4). We thus hypothesize that *eRank collapse* is a primary mechanism underlying *reasoning collapse* of width-reduced EDistilled LLMs, which necessitates a theoretical investigation into the optimization dynamics of projection-based width reduction.

### 4.1. Gradient-driven ERank Collapse

To understand the origin of *eRank collapse* under width reduction, we analyze the optimization dynamics induced by projection-based compression using a linear autoencoder model (Wang et al., 2016) as a theoretical proxy for the teacher-student representation alignment process.

**Problem Setup.** Let $X \in \mathbb{R}^{L \times D}$ denote a zero-centered and row-normalized representation matrix. Width-reduced EDistill introduces a down-projection matrix $Q \in \mathbb{R}^{D \times D'}$ and an up-projection matrix $O \in \mathbb{R}^{D' \times D}$ ($D' < D$), optimized to reconstruct $X$ via $\mathcal{L} = \frac{1}{2L} \|X - XQO\|_F^2$.

Following Saxe et al. (2013); Arora et al. (2019); Jing et al. (2021), we model the training process with a small learning rate as continuous-time gradient flow:

**Lemma 4.1** (Evolution of Projection Matrices). *Under the reconstruction loss $\mathcal{L}$, the gradient flow dynamics for the projection matrices $O$ and $Q$ are governed by the following system of Ordinary Differential Equations:*

$$\dot{O} = -\nabla_O \mathcal{L} = -Q^\top \Sigma (QO - I), \qquad (5)$$

$$\dot{Q} = -\nabla_Q \mathcal{L} = -\Sigma (QO - I) O^\top, \qquad (6)$$

*where $\dot{O}$ and $\dot{Q}$ denote the time-derivatives $\frac{dO}{dt}$ and $\frac{dQ}{dt}$ respectively, $\Sigma = \frac{1}{L} X^\top X$ is the covariance matrix and $I$ is the identity matrix of the corresponding dimensions.*

A key property of this system is that the difference between the Gram matrices of $Q^\top$ and $O$ remains invariant over time, which we formalize in Lemma D.4. Consequently, if the matrices are initialized such that $Q(0)^\top Q(0) = O(0)O(0)^\top$, they remain "balanced" throughout the entire optimization process, i.e., $Q(t)^\top Q(t) = O(t)O(t)^\top$ for all $t$.

In practice, existing width-reduced EDistill methods employ small random initialization (a normal distribution with mean = 0.0, std = 0.02). Under such conditions, $Q(0)^\top Q(0) \approx O(0)O(0)^\top \approx 0$. Then we define the product matrix $M = QO \in \mathbb{R}^{D \times D}$ and the Singular Value Decomposition (SVD) of $M$ as $M = U_M E_M V_M^\top$, where $E_M = \text{diag}(\sigma_M^1, \ldots, \sigma_M^{D'})$. It holds that:

**Theorem 4.2** (Singular Value Dynamics). *Given the balanced condition $Q^\top Q = OO^\top$, the evolution of the $r$-th singular value $\sigma_M^r$ of the product matrix $M = QO$ is:*

$$\dot{\sigma}_M^r = 2\sigma_M^r (u_M^r)^\top \Sigma (v_M^r - \sigma_M^r u_M^r). \qquad (7)$$

*where $u_M^r$ and $v_M^r$ are the $r$-th left and right singular vectors of $M$, respectively.*

**Mechanism of ERank Collapse.** Theorem 4.2 reveals a critical "winner-take-all" dynamic. During the early stages of training with small random initialization ($\sigma_M^r \ll 1$), the term $(v_M^r - \sigma_M^r u_M^r)$ is approximately $v_M^r$. Consequently, the growth rate $\dot{\sigma}_M^r$ is proportional to $\sigma_M^r$ itself, leading to exponential divergence. This implies that singular values associated with the principal components of $\Sigma$ (where $(u_M^r)^\top \Sigma v_M^r$ is large) will escalate rapidly and dominate the representation space. Conversely, other singular values remain trapped near zero. According to Lemma D.10, the singular values of the projection matrix $Q$ are the square roots of those of $M$. Therefore, this highly skewed distribution in $M$ directly propagates to $Q$, leading to the *eRank collapse* of the compressed representation $XQ$, where the model over-focuses on a few dominant feature directions while discarding the rest. Detailed proofs of Lemma and Theorem are provided in Appendix D.1 to D.4.

### 4.2. Consequences of ERank Collapse

After explaining why projection-based width reduction yields *eRank collapse*, we next formalize its consequences by bounding how low eRank collapses inter-token distinguishability in the output probability space, undermining multi-step reasoning ability. To be specific, we employ the Total Variation (TV) distance to quantify the distinguishability between token predictions:

**Theorem 4.3** (Probability Distinguishability Bound). *Let $X^{l_1}, X^{l_2} \in \mathbb{R}^D$ be token features from a row-normalized, zero-centered matrix $X \in \mathbb{R}^{L \times D}$. The output distribution is $P^l = \text{Softmax}(Z^l)$, where $Z^l = \text{Norm}_{\text{final}}(X^l)W_u$ are logits and $g_{\text{final}}$ is the scaling parameter of $\text{Norm}_{\text{final}}$. The minimum TV distance is bounded by:*

$$\min_{l_1 \neq l_2} d_{TV}(P^{l_1}, P^{l_2}) \leq \frac{\sqrt{\text{Voc}}}{2} \|\text{diag}(g_{\text{final}})W_u\|_2$$

$$\cdot \sqrt{2 - 2\sqrt{\frac{1}{L-1} \left(\frac{L}{\text{eRank}(X)} - 1\right)}}, \qquad (8)$$

*where $\|\text{diag}(g_{\text{final}})W_u\|_2$ is the spectral norm of the scaled unembedding matrix.*

Theorem 4.3 reveals that when $\mathrm{eRank}(X)$ is low, the model tends to produce nearly identical output probability distributions for distinct tokens. This indistinguishability undermines reasoning by blurring (i) key input variables/conditions in the prompt, leading to errors and hallucinations (Mamidanna et al., 2025), and (ii) successive reasoning steps in multi-step generation, causing repetitive or degenerate trajectories (Ye et al., 2024; Tan et al., 2025; Fu et al., 2026). In the extreme case, *eRank collapse* makes the model completely degenerate:

> **Theorem 4.4** (Binary State Collapse). *Assume the representation matrix $X \in \mathbb{R}^{L \times D}$ is zero-centered and row-normalized. If $\mathrm{eRank}(X) = 1$, the rank of $X$ becomes 1. For any $l$-th token, its representation $X^l$ satisfies $X^l = X^1$ or $X^l = -X^1$. Consequently, the corresponding logits satisfy $Z^l = Z^1$ or $Z^l = \mathrm{Norm}_{\mathrm{final}}(-X^1)W_{\mathrm{u}}$.*

Proofs of Theorem D.13, 4.3, and 4.4 are provided in Appendix D.5, D.6, D.7.

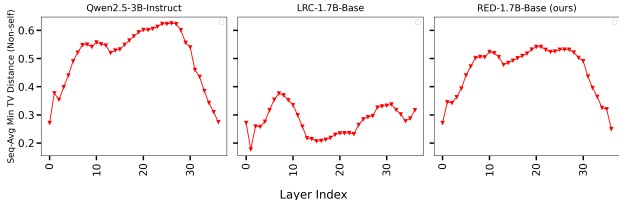

*Figure 3.* Minimum TV Divergence (non-self) between token output probability distributions across layers.

**Experimental Validation.** To empirically validate our theoretical findings, we perform a layer-wise analysis on sequences sampled from the Nemotron-Pretraining-Dataset, extracting hidden representations to compute their corresponding output probability distributions. As illustrated in Figure 3, LRC-1.7B-Base (middle) exhibits a significantly smaller minimum TV divergence than the teacher model (left). Furthermore, the Pearson correlation (Sedgwick, 2012) between eRank and minimum TV distance across all layers of the three models is **0.74**. These results are consistent with Theorem D.6, confirming that low eRank yields indistinguishable token output probability distributions.

## 5. RED: Reasoning-preserved Efficient Distillation for LLMs

Building on previous analyses, we propose **R**easoning-preserved **E**fficient **D**istillation (**RED**) for LLMs. This simple method preserves the multi-step reasoning ability of LLMs via *Activation-aware Initialization* that stabilizes early optimization and mitigates *eRank collapse* of hidden representations by initializing projection matrices as channel-selection matrices.

**Activation-aware Initialization.** Instead of small random values, we initialize the down-projection matrix $Q \in$

$\mathbb{R}^{D \times D'}$ as a channel-selection matrix $H$. Let $G = \{G_1, \ldots, G_{D'}\} \subset \{1, \ldots, D\}$ denote the indices of the $D'$ most important activation channels. We define $Q(0) = H$, where $H_{ij} = 1$ if $i = G_j$ and 0 otherwise. For the up-projection matrix $O \in \mathbb{R}^{D' \times D}$, we set $O(0) = H^\top$, such that $Q(0)^\top = O(0)$. This initialization yields two key advantages: (i) **Residual Preservation.** By employing the same selection matrix $H$ for all projection matrices, the student preserves the selected activation subspace at initialization, resulting in a stable, teacher-aligned residual structure. (ii) **Spectral Stability.** The product matrix $M(0) = Q(0)O(0)$ is a symmetric projection matrix whose initial dynamics are characterized by:

> **Theorem 5.1** (Vanishing Initial Dynamics). *Under the activation-aware initialization, at $t = 0$, the singular values of $M(0)$ satisfy $\sigma_M^r \in \{0, 1\}$, and their initial derivatives vanish, i.e., $\dot{\sigma}_M^r = 0$ for all $r \in \{1, \ldots, D\}$.*

Theorem 5.1 (Proof in Appendix D.8) implies that the singular values remain stable at initialization. This mitigates the "winner-take-all" effect in random initializations, thereby safeguarding the model against *eRank collapse*.

**Activation-Based Importance Estimation.** To construct the index set $G$, we employ activation-based importance estimation commonly used in structured pruning (Muralidharan et al., 2024). We estimate the importance of each channel of activations using a small calibration dataset $\mathcal{D}_{\mathrm{pre}}$. Let $X_i^{(k)} \in \mathbb{R}^{L \times D}$ denote the output hidden representation matrix of the $k$-th sequence in the $i$-th Transformer layer ($X_0^{(k)}$ is the output of the embedding layer). We represent the $l$-th token as $X_i^{l,(k)} \in \mathbb{R}^D$ and its $j$-th channel value as $(X_i^{l,(k)})_j$. First, we compute the mean absolute activation for the $j$-th channel of layer $i$ for this specific sequence:

$$s_{i,j}^{(k)} = \frac{1}{L} \sum_{l=1}^{L} \left| (X_i^{l,(k)})_j \right|. \tag{9}$$

To aggregate the importance across the entire dataset, the global importance score for the $j$-th channel, $\bar{s}_j$, is obtained by averaging the $L_2$ norm of $s_{i,j}^{(k)}$ over all sequences across $N$ Transformer blocks and one embedding layer:

$$\bar{s}_j = \frac{1}{N+1} \sum_{i=0}^{N} \sqrt{\sum_{k=1}^{|\mathcal{D}_{\mathrm{pre}}|} \left( s_{i,j}^{(k)} \right)^2}. \tag{10}$$

Finally, we obtain the set of important channel indices $G = \mathrm{Argmax}(\bar{s}, D')$, which consists of the indices corresponding to the $D'$ largest values in the global importance vector $\bar{s} = [\bar{s}_1, \ldots, \bar{s}_D]$. Notably, we further introduce various activation-based importance estimation strategies. These strategies demonstrate consistent performance when employed in RED (Appendix G.6), highlighting the robustness of our framework.

**Experimental Validation.** We validate the proposed initialization strategy using a linear autoencoder proxy trained on hidden representations from the 26th layer of Qwen2.5-3B-Instruct. We collect representations using 15 samples from SlimOrca, forming $X \in \mathbb{R}^{4169 \times 2048}$ (eRank 290), and train an autoencoder with a bottleneck dimension $D' = 1200$ for 100 epochs using an Adam optimizer with a learning rate $1 \times 10^{-4}$. As shown in Table 3, random initialization leads to severe *eRank collapse*, with low eRank for hidden (81.3) and reconstructed (67.06) representations. Activation-aware initialization maintains substantially higher eRank and achieves lower reconstruction loss, confirming its effectiveness in mitigating *eRank collapse*. In practice, we further visualize the relative singular values (i.e., all singular values divided by the maximum one) of the learnable projection matrices during the training of LRC and RED. As shown in Figure 4, the singular value space of RED stabilizes in an even distribution, whereas LRC exhibits some relatively small singular values during training. Furthermore, Theorem 4.2 predicts a winner-take-all dynamic in the early stages of training, namely $\dot{\sigma}_M^r \propto \sigma_M^r$. While the exact divergence rate is difficult to predict in the full nonlinear network, we can directly test the core prediction of the theory: whether the empirical growth rate ($\Delta\sigma$) is positively correlated with the singular value ($\sigma$) itself in early training. As summarized in Table 2, in the early phase (20k–40k), the correlation is strongly positive ($r = 0.8948$ for the Up projection). This provides quantitative support for our theoretical prediction that larger singular values grow proportionally faster early on, driving rapid spectral imbalance from small random initialization. As training progresses and singular values grow large, architectural bounds (e.g., RMSNorm, SwiGLU) and the loss function limit further divergence, which eventually turns the correlation negative. However, much of the geometric damage to the representation space (*eRank collapse*) has already been incurred during the early phase.

*Table 2.* Pearson correlation ($r$) between empirical singular values ($\sigma$) and their growth ($\Delta\sigma$) across different training intervals for projection matrices (FFN module of the 20-th layer) in LRC-1.5B.

| Interval | Up $r$ | Down $r$ | Phase |
|---|---|---|---|
| 20k → 40k | **0.8948** | **0.5454** | Early |
| 60k → 80k | 0.0057 | -0.6936 | Transition |
| 180k → 200k | -0.6703 | -0.9388 | Late |

# 6. Numerical Experiments

**Training Configuration.** Apart from adjusting the KL divergence temperature from 40 to 1 (following Sreenivas et al. (2024); Muralidharan et al. (2024)), our training configuration remains identical to the settings in Hao et al. (2025). Specifically, we train a series of RED models (RED-1.5/1.7/2.7/4B) using strong open-source Instruct versions of LLMs as teachers (Dubey et al., 2024; Team et al., 2024).

*Table 3.* Initialization impact on reconstruction Loss and eRank.

| Initialization | Recon. Loss↓ | Hidden eRank↑ | Recon. eRank↑ |
|---|---|---|---|
| Random | 1.24 | 81.30 | 67.06 |
| **Activation-aware (Ours)** | **0.08** | **263.28** | **249.44** |

*Figure 4.* Relative singular values of projection matrices, aligning the FFN module of the 20th layer of LRC and RED-1.5B.

All models are trained with packed sequences of length 2,048 for computational efficiency. We employ the Adam optimizer with $\beta_1 = 0.9$ and $\beta_2 = 0.999$. Training runs on 8 NVIDIA H800 GPUs using PyTorch, the transformers package (Wolf et al., 2019), and deepspeed (Aminabadi et al., 2022) for distributed parallelism. It is worth noting that the learnable projection matrices inserted into each module across all layers in both RED and LRC are trained separately. The only difference is that in LRC, these matrices are randomly initialized, whereas in RED, they are initialized with specific values to preserve the global substructure. Details of implementation in Appendix C. The hyperparameter settings and model configurations are provided in Appendix F.1 and F.3. When employing the Base model as the teacher model, RED still demonstrated a significant advantage (Appendix G.3).

**Calibration Datasets.** Consistent with Muralidharan et al. (2024), we require the calibration dataset $\mathcal{D}_{pre}$ to estimate the importance of each channel within the hidden representations, thereby enabling the initialization of projection matrices. To be specific, we sampled five examples from each of the three subsets "Nemotron-CC-Diverse-QA", "Nemotron-SFT-Code", and "Nemotron-SFT-General" within the Nemotron-Pretraining-Dataset(Adler et al., 2024) (15 samples in total) to estimate the importance. Compared to the training dataset, the calibration dataset is significantly smaller in volume and computationally less demanding (typically taking less than three minutes on a single GPU). In Appendix G.2, we demonstrate that the choice of dataset and the number of samples drawn have a negligible impact on the final selection of important channels, highlighting the robustness of our approach.

**Training Datasets.** We construct the retraining corpus $\mathcal{D}_{\text{post}}$ by blending data from Fineweb-Edu(Penedo et al., 2024), DCLM(Li et al., 2024), OpenHermes(Teknium, 2023) and CosmopediaV2(Allal et al., 2025), consistent with Hao et al. (2025). The combined pre-training dataset is randomly shuffled. Details are in Appendix F.2. It is

*Table 4.* **Main Results on ∼2B Series.** Models are grouped by: Full-parameter and EDistill . RED series (Ours) achieves SOTA in both **Gen. Avg.** and **Reas. Avg.** Compared to the Full-parameter method, it requires far fewer tokens and computational resources.

| Paradigm | Compute-Intensive Full-parameter | | | | | EDistill | | | | | |
|---|---|---|---|---|---|---|---|---|---|---|---|
| Model (Size) | Llama3.2 (1.2B) | InternLM2 (1.8B) | SmolLM2 (1.7B) | MiniCPM (1.0B) | Gemma3 (1.0B) | LRC (1.5B) | RED (Ours) (1.5B) | LRC (1.7B) | RED (Ours) (1.7B) | LRC (2.7B) | RED (Ours) (2.7B) |
| Teacher | Llama3.1-8B | – | – | – | – | Llama3.2-3B | Llama3.2-3B | Qwen2.5-3B | Qwen2.5-3B | Llama2-7B | Llama2-7B |
| # Tokens | 9T | 2T | 11T | 1T | 2T | 10B | 10B | 20B | 20B | 10B | 10B |
| Dataset | N/A | N/A | SmolLM | N/A | N/A | Mixed-1.1 | Mixed-1.1 | Mixed-1.1 | Mixed-1.1 | RedPajama | RedPajama |
| ▼ *General Ability* ↑ | | | | | | | | | | | |
| MMLU | 0.37 | 0.41 | 0.48 | 0.44 | 0.24 | 0.50 | 0.51 | 0.55 | 0.55 | 0.32 | 0.32 |
| PIQA | 0.78 | 0.80 | 0.77 | 0.76 | 0.75 | 0.72 | 0.74 | 0.73 | 0.74 | 0.66 | 0.75 |
| ARC-E | 0.61 | 0.70 | 0.74 | 0.67 | 0.72 | 0.70 | 0.69 | 0.65 | 0.72 | 0.55 | 0.64 |
| ARC-C | 0.36 | 0.38 | 0.47 | 0.40 | 0.38 | 0.42 | 0.44 | 0.43 | 0.45 | 0.29 | 0.33 |
| BoolQ | 0.64 | 0.70 | 0.72 | 0.69 | 0.66 | 0.73 | 0.76 | 0.76 | 0.80 | 0.75 | 0.66 |
| WinoGrande | 0.60 | 0.70 | 0.65 | 0.65 | 0.59 | 0.62 | 0.64 | 0.63 | 0.65 | 0.63 | 0.65 |
| HellaSwag | 0.64 | 0.65 | 0.71 | 0.61 | 0.62 | 0.56 | 0.62 | 0.59 | 0.65 | 0.65 | 0.69 |
| TruthfulQA | 0.38 | 0.40 | 0.37 | 0.37 | 0.37 | 0.48 | 0.47 | 0.53 | 0.53 | 0.42 | 0.40 |
| Gen. Avg. | 0.55 | 0.59 | 0.61 | 0.57 | 0.54 | 0.59 | 0.61 | 0.61 | **0.64** | 0.53 | 0.56 |
| ▼ *Multi-step Reasoning (Math & Code)* ↑ | | | | | | | | | | | |
| GSM8K | 0.07 | 0.23 | 0.30 | 0.39 | 0.03 | 0.21 | 0.44 | 0.09 | 0.42 | 0.09 | 0.15 |
| MBPP | 0.27 | 0.25 | 0.34 | 0.32 | 0.09 | 0.07 | 0.21 | 0.09 | 0.16 | 0.07 | 0.14 |
| HumanEval | 0.18 | 0.01 | 0.01 | 0.06 | 0.06 | 0.00 | 0.19 | 0.01 | 0.16 | 0.01 | 0.01 |
| Reas. Avg. | 0.17 | 0.16 | 0.22 | 0.26 | 0.06 | 0.09 | **0.28** | 0.06 | 0.25 | 0.06 | 0.10 |

worth noting that for SliceGPT and LLMPruner, which lack open-source checkpoints, we utilized data from their official implementations. Similar to LRC, we extract 10B tokens from RedPajama to train RED-2.7B, employing Llama2-7B-Chat as the teacher model.

**Baselines.** To comprehensively evaluate the efficacy of our method, we compare it against a diverse set of representative baselines: Compute-Intensive Full-parameter Models: models derived from massive retraining or pretraining, representing the performance upper bound under high-resource settings (trillions of tokens and extensive GPU hours). We include *pruning-then-retraining* methods such as Minitron (Muralidharan et al., 2024) and Llama-3.2-1B/3B (Dubey et al., 2024), as well as SOTA *from-scratch* SLMs including MiniCPM (Hu et al., 2024a), SmolLM2 (Allal et al., 2025), Gemma3 (Team et al., 2025), and InternLM2 (Cai et al., 2024). LoRA-style Recovery: SliceGPT (Ashkboos et al., 2024) and LLM-Pruner (Ma et al., 2023). EDistill: *depth-reduced* LLMStreamline (Chen et al., 2024), and *width-reduced* LRC (Hao et al., 2025). It is worth noting that our baseline and RED series are all Base model. RED models also outperform the Instruct versions of many SLMs (Appendix G.4). As there are no checkpoints available, SliceGPT, LLMPruner, and LRC-2.7B are trained based on the official implementation. Reported Full-parameter SLMs, the LRC series, and LLMStreamline are from HuggingFace (checkpoints in Appendix F.4).

**Evaluation protocols.** To ensure a fair and reproducible comparison, all evaluations are conducted using the lm-evaluation-harness framework (Gao et al., 2024), leverag-

ing the transformers package (Wolf et al., 2019) as the inference backend. We evaluate our model across a suite of benchmarks, categorized by the core ability they assess: (i) **General Ability**: Multiple-choice tasks targeting: (1) *Scientific Understanding and Reading Comprehension*: ARC-E/C (Clark et al., 2018) and BoolQ (Clark et al., 2019); (2) *Commonsense Understanding*: PIQA (Bisk et al., 2020), WinoGrande (Sakaguchi et al., 2021), and HellaSwag (Zellers et al., 2019); and (3) *World Knowledge & Truthfulness*: MMLU (Hendrycks et al., 2020) and TruthfulQA (Lin et al., 2022). (ii) **Multi-step Reasoning**: For tasks requiring complex logical execution and generation, we evaluate: (1) *Mathematical Reasoning*: GSM8K (Cobbe et al., 2021), (2) *Code Generation*: HumanEval (Chen, 2021) and MBPP (Austin et al., 2021). Detailed settings for each benchmark are provided in Appendix F.5.

**Superiority at the 2B Scale.** As shown in Table 4, while the SOTA width-reduction Edistill method LRC suffers from severe performance degradation in multi-step reasoning (e.g., dropping to 0.06–0.09 on Reas. Avg.), RED effectively recovers these capabilities to 0.10–0.28, representing a ∼2× to 4× improvement across all teacher models. When compared to Full-parameter pretraining or retraining SLMs such as Gemma3-1B, Llama3.2-1B, and SmolLM2-1.7B, RED achieves SOTA in both *Gen. Avg.* and *Reas. Avg..* Crucially, RED achieves these results with a fraction of the data budget, using only 10B–20B tokens compared to the trillions of tokens required for Full-parameter training. This confirms that our method allows the model to inherit the ability from the teacher more efficiently.

*Table 5.* **Main results on the ∼4B family.** Paradigm classification: Full-Parameter , LoRA-style , and EDistill . All models are evaluated with the same setup (see Appendix F.5). Despite being trained with only 18B tokens on the Mixed-2.0 corpus, RED achieves state-of-the-art multi-step reasoning performance among EDistilled methods, and remains competitive with compute-intensive full-parameter baselines that use substantially larger data / compute budgets (often including large-scale code and math corpora).

| Paradigm | Full-Parameter | | | LoRA-style | | EDistill | | | |
|---|---|---|---|---|---|---|---|---|---|
| Model (Size) | Llama3.2 (3.0B) | Gemma3 (4.0B) | Minitron (4.0B) | LLM-Pruner (4.7B) | SliceGPT (4.7B) | LLMStr. (4.7B) | LLMStr. (5.4B) | LRC (4.0B) | RED (Ours) (4.0B) |
| Teacher | Llama3.1-8/70B | – | Minitron-15B | Llama2-7B | Llama2-7B | Llama2-7B | Llama3.1-8B | Qwen2.5-7B | Qwen2.5-7B |
| # Tokens | 9T | 4T | 94B | 50M | 50M | 60M | 1.3B | 18B | 18B |
| Dataset | N/A | N/A | N/A | Alpaca | Alpaca | SlimPajama | SlimPajama | Mixed-2.0 | Mixed-2.0 |
| ▼ *General Ability* ↑ | | | | | | | | | |
| MMLU | 0.54 | 0.58 | 0.56 | 0.23 | 0.26 | 0.31 | 0.62 | 0.65 | 0.62 |
| PIQA | 0.78 | 0.80 | 0.78 | 0.72 | 0.68 | 0.73 | 0.76 | 0.76 | 0.79 |
| ARC-E | 0.72 | 0.82 | 0.76 | 0.51 | 0.53 | 0.63 | 0.71 | 0.75 | 0.77 |
| ARC-C | 0.46 | 0.55 | 0.45 | 0.25 | 0.27 | 0.35 | 0.43 | 0.52 | 0.53 |
| BoolQ | 0.73 | 0.79 | 0.72 | 0.65 | 0.66 | 0.75 | 0.78 | 0.85 | 0.84 |
| WinoGrande | 0.70 | 0.70 | 0.68 | 0.55 | 0.58 | 0.66 | 0.71 | 0.68 | 0.71 |
| HellaSwag | 0.74 | 0.62 | 0.72 | 0.58 | 0.51 | 0.67 | 0.69 | 0.71 | 0.75 |
| TruthfulQA | 0.39 | 0.40 | 0.43 | 0.38 | 0.37 | 0.41 | 0.45 | 0.56 | 0.54 |
| **Gen. Avg.** | 0.63 | 0.66 | 0.64 | 0.48 | 0.48 | 0.56 | 0.64 | **0.69** | **0.69** |
| ▼ *Multi-step Reasoning (Math & Code)* ↑ | | | | | | | | | |
| GSM8K | 0.25 | 0.38 | 0.25 | 0.01 | 0.03 | 0.03 | 0.22 | 0.34 | **0.49** |
| MBPP | 0.38 | 0.47 | 0.35 | 0.02 | 0.04 | 0.11 | 0.23 | 0.14 | **0.28** |
| HumanEval | 0.27 | 0.35 | 0.04 | 0.00 | 0.01 | 0.00 | 0.16 | 0.07 | **0.23** |
| **Reas. Avg.** | 0.30 | 0.40 | 0.21 | 0.01 | 0.03 | 0.05 | 0.20 | 0.18 | **0.33** |

**Scalability at the 4B Scale.** The advantages of RED remain robust as the model scales to 4B parameters. Compared to Full-parameter *pruning-then-retraining* methods like Minitron-4B and Llama-3.2-3B, which involve extensive high-resource retraining, RED maintains a significant edge. Notably, Gemma3-4B exhibits a significant leap in reasoning performance compared to its 2B version. This is likely attributed to the massive scale of its proprietary pre-training dataset (4T tokens), which is heavily enriched with math and code corpora. In contrast, RED-4B is distilled using only 18B tokens from educational datasets. Despite this 200× difference in data volume, RED-4B still achieves a competitive *Reas. Avg.* of 0.33 and a superior *Gen. Avg.* of 0.69. These results highlight that RED provides a data-efficient alternative to massive data scaling.

**Efficiency.** To quantify the computational efficiency, we take RED-1.5B (comprising 0.94B trainable parameters) as a representative case. When training on a single node equipped with 8×H800 GPUs, our method consumes a peak memory of 56 GB per GPU with a per-card batch size of 6,144 tokens. The entire training process converges within approximately 30 hours, utilizing a total data budget of only 10B tokens. In stark contrast, Full-parameter retraining baselines such as Minitron-4B and Llama-3.2-3B require significantly more computational resources, often involving 256 or even thousands of A100/H100 GPUs, and are trained on much larger datasets ranging from 94B to 9T tokens. This comparison highlights that our approach achieves a high

level of performance with significantly less compute and data overhead. In Appendix G.1, we demonstrate that the multi-step reasoning ability of RED exhibits significantly faster recovery speed during training compared to LRC.

*Table 6.* Impact of distillation temperature $\tau$ on general ability (MMLU) and multi-step reasoning (GSM8K). To ensure fairness, models are trained under identical settings. **Bold** indicates the best performance for each method.

| Method | Temprature | General (MMLU) ↑ | Reasoning (GSM8K) ↑ |
|---|---|---|---|
| LRC-1.5B | $\tau = 1$ | 0.25 | 0.05 |
| | $\tau = 40$ | 0.50 | 0.04 |
| RED-1.5B (Ours) | $\tau = 1$ | 0.50 | **0.44** |
| | $\tau = 40$ | **0.52** | 0.08 |

**Sensitivity to Distillation Temperature.** We study the effect of the distillation temperature $\tau$ on RED-1.5B and LRC-1.5B under identical training settings, with results summarized in Table 6. LRC-1.5B fails to converge at a low temperature ($\tau = 1$) and only becomes trainable at a high temperature ($\tau = 40$), which smooths the teacher distribution. We attribute this behavior to its randomly initialized projection matrices, which disrupt the student's internal structure and hinder alignment with the teacher under strict (low-temperature) supervision. In contrast, RED-1.5B achieves both its best general performance and its peak multi-step reasoning performance at $\tau = 1$. Specifically, RED-1.5B attains a GSM8K accuracy of 0.44 at $\tau = 1$, whereas increasing the temperature to $\tau = 40$ leads to a severe degradation in reasoning performance (dropping to 0.08). This sharp contrast indicates that multi-step reasoning

relies on high-precision "dark knowledge" encoded in the teacher's distribution, which is diluted by high-temperature smoothing. By preserving spectral stability and residual structure through activation-aware initialization, RED enables effective alignment at low temperature and captures this sharp reasoning-relevant information without relying on over-smoothed distillation targets.

**Comparison of Orthogonalization Techniques.** Orthogonalization techniques are widely used in self-supervised learning and knowledge distillation to prevent representation collapse in parameter matrices (He et al., 2024; Miles et al., 2024). In this section, we compare these generic techniques with our proposed method in both a linear autoencoder proxy and a full distillation setup. Specifically, we evaluate two common baselines: (1) *Random Orthogonal Initialization* (Miles et al., 2024), which initializes the learnable projection matrix as a random orthogonal matrix, and (2) *Orthogonal Regularization (OR)* (He et al., 2024), which applies an $L_2$ penalty on projection matrices to encourage orthogonality during training. The results are summarized in Table 7. While generic orthogonal initialization and regularization improve the eRank compared to standard random initialization, they still start from random singular directions. Consequently, they significantly underperform RED in both reconstruction loss and downstream reasoning tasks. This clarifies a critical distinction between RED and generic orthogonal methods: **RED not only encourages a healthier singular value spectrum but also explicitly preserves important teacher subspaces at initialization.**

*Table 7.* Comparison of orthogonalization techniques. **Top:** Linear autoencoder proxy. **Bottom:** Edistill setup evaluating downstream performance on MMLU and GSM8K.

| Method (Autoencoder) | Recon. ↓ | eRank ↑ |
|---|---|---|
| Random | 1.24 | 67.06 |
| Random + OR (0.001) | 1.24 | 111.82 |
| Random + OR (0.01) | 1.52 | 172.60 |
| Random Orth. Init | 0.50 | 149.97 |
| Ours (RED) | **0.08** | **263.28** |

| Method (Edistll) | MMLU | GSM8K |
|---|---|---|
| LRC (Random) | 0.50 | 0.21 |
| LRC + Random Orth. Init | 0.50 | 0.30 |
| LRC + OR (1e-4) | 0.48 | 0.26 |
| RED (Ours) | **0.51** | **0.44** |

**Impact of eRank Collapse on Multi-step Reasoning Reasoning.** Classification tasks (e.g., MMLU) primarily require preserving the Top-1 ranking among few candidates, which is robust to blurry representations. In contrast, multi-step reasoning requires distinguishing and updating intermediate states over a long trajectory; collapsed eRank makes these states inseparable, causing cascading errors. We demonstrate this via a GSM8K case study.

---

**Prompt:** A tree grows 4 inches every month. If the tree is currently 5 feet tall, how tall will it be in 2 years? (1 foot = 12 inches)

---

**RED (Ours, Succeeded):** The tree is currently $5 \times 12 = \mathbf{60}$ inches tall. The tree grows 4 inches every month. So, in 1 year, the tree will grow $4 \times 12 = 48$ inches. In 2 years, the tree will grow $48 \times 2 = \mathbf{96}$ inches. The tree will be $\mathbf{60} + \mathbf{96} = \mathbf{156}$ inches tall.

**LRC (Failed):** The tree is currently $5 \times 12 = \mathbf{60}$ inches tall. In 1 year, the tree will grow $4 \times 12 = 48$ inches. In 2 years, the tree will grow $48 + 48 = \mathbf{96}$ inches. So, the tree will be $\mathbf{96} \times \mathbf{12} = \mathbf{1,192}$ inches tall.

---

As shown above, RED preserves key states (60 and 96), whereas LRC loses the initial height (60) and wrongly converts 96 again. We trace this failure to three internal phenomena aligning with our theory: **(1) Representation Indistinguishability:** The average cosine similarity of hidden representations across reasoning steps is **high for LRC (0.40)** but **low for RED (0.13)**. Collapsed eRank forces distinct reasoning steps into overlapping subspaces, making them hard to distinguish. **(2) Loss of Crucial Context:** For the critical token "*be*" (in "*the tree will be*"), LRC's Top-5 nearest tokens in representation space (48, 12, 4) **completely miss the crucial initial state 60**. Conversely, **RED successfully retrieves 60** in its Top-5. **(3) High Output Entropy:** At the final operator decision (target: +), LRC yields a high entropy of **1.28** (Top-3 probs: * 50%, / 20%, + 20%), confirming that eRank collapse blurs output token distinctions. In contrast, RED exhibits a sharp entropy of **0.18**, confidently predicting **+ (97%)**.

## 7. Conclusion and Discussion

In this work, we identify and systematically investigate *reasoning collapse* in EDistilled LLMs, where the multi-step reasoning ability severely degrades. Through theoretical analysis, we reveal how the singular values of randomly initialized projection matrices become unevenly distributed, leading to eRank collapse and thus token indistinguishability, undermining the multi-step reasoning ability. To resolve this issue, we introduce **RED**, a reasoning-preserved efficient distillation framework utilizing *activation-aware initialization*. Our theoretical analysis and extensive experiments on Llama and Qwen series demonstrate that our RED effectively mitigates eRank collapse and maintains strong multi-step reasoning ability while preserving SOTA general ability with high training efficiency.

**Future Work.** We aim to extend the RED framework to distilling strong reasoning models (e.g., DeepSeek-R1 series (Guo et al., 2025)). Another compelling direction is the adaptation of RED to Multimodal Large Language Models and Vision-Language-Action models (Wang et al., 2023; Xu et al., 2025). These architectures, often constrained by high inference latency in embodied AI and real-time vision tasks.

## Acknowledgements

The work described in this paper is supported by Research Grants Council of the Hong Kong Special Administrative Region, China (Project No. PolyU/15206322 and PolyU/15227424); Otto Poon Charitable Foundation Smart Cities Research Institute, The Hong Kong Polytechnic University (CD06); Otto Poon Research Institute for Climate-Resilient Infrastructure, The Hong Kong Polytechnic University (ZH8U); Research Centre for Digital Transformation of Tourism, The Hong Kong Polytechnic University (BBGU); and International Centre of Urban Energy Nexus (CE0G). The contents of this article reflect the views of the authors, who are responsible for the facts and accuracy of the information presented herein.

## Impact Statement

This work aims to facilitate the seamless transition of frontier AI to the edge by enabling the efficient development of compact, high-performance language models, which require significantly fewer computational resources for deployment. By introducing activation-aware initialization for projection matrices within the Efficient Distillation (EDistill) framework, we enable the compression of large-scale teacher models into compact student models using standard distillation objectives, which not only maintains high training efficiency but also mitigates the "reasoning collapse" prevalent in existing methods, thus providing the community with more robust and capable Base models. We believe this research contributes to the democratization of AI by making high-performing models accessible on resource-constrained devices. Our study utilizes exclusively publicly available models and educational datasets, and we do not foresee any direct negative societal impacts arising from this work.

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

# Appendix

**Table of Contents**

## A. Limitations

Despite the promising results, our study has several limitations that warrant further exploration. First, our distillation process primarily utilized general-purpose corpora; the specific impact of integrating specialized mathematical or code-heavy datasets on the reasoning-efficiency trade-off remains to be systematically quantified. Furthermore, while RED provides a robust *base* for reasoning, we have not yet evaluated the synergistic effects of post-training techniques, such as Reinforcement Learning or instruction-tuning on specialized datasets, in further unlocking the ability of RED-distilled models.

## B. Distillation Objectives

In this section, we introduce common distillation objectives, which align the smaller student model with the teacher model. Let $X_i$ and $X_j' \in \mathbb{R}^{L \times D}$ denote the hidden representations of the $i$-th layer in the teacher and the $j$-th layer in the student, respectively. Similarly, let $Z, Z' \in \mathbb{R}^{L \times V}$ represent their output logits as defined in Eq. (2) and $Z^l, (Z^l)' \in \mathbb{R}^V$ denote the logit vectors of the $l$-th token for the teacher and student.

A primary objective is *representation alignment*, which minimizes the discrepancy between intermediate hidden representations. A representative formulation utilizes the Mean Squared Error (MSE) with the Frobenius norm:

$$\mathcal{L}_{\text{REP}} = \frac{1}{L} \|X_i - X_j'\|_F^2. \tag{11}$$

Another fundamental objective is the *language modeling loss* (i.e., next token prediction). Given the ground truth token sequence $y = \{y_1, \dots, y_L\}$, the student model is trained to minimize the standard negative log-likelihood:

$$\mathcal{L}_{\text{LM}} = -\frac{1}{L} \sum_{l=1}^{L} \log \left( \frac{\exp((Z^l)'_{y_l})}{\sum_{k=1}^{V} \exp((Z^l)'_k)} \right), \tag{12}$$

where $(Z^l)'_{y_l}$ denotes the logit value corresponding to the ground truth token $y_l$ at position $l$.

Furthermore, *logit distillation* is frequently adopted to transfer the detailed probability distributions ("dark knowledge") from the teacher. This is formally expressed using the Kullback-Leibler (KL) divergence between temperature-scaled soft targets:

$$\mathcal{L}_{\text{KL}} = \frac{1}{L} \sum_{l=1}^{L} \text{KL} \left( \text{softmax} \left( \frac{Z^l}{\tau} \right) \,\Big\|\, \text{softmax} \left( \frac{(Z^l)'}{\tau} \right) \right), \tag{13}$$

where $\tau$ is a temperature coefficient. Existing methods may employ these objectives individually or in a weighted combination.

## C. Implementation Details of RED

In this section, we provide the specific implementation details of RED. Following the architectural formulation in Section 3 and the width-reduced paradigm, we detail how the projection matrices are integrated and trained.

For each Transformer layer $i$ in the teacher LLM, we identify the key weight matrices that define the hidden state transformations. Specifically, for the Llama-style architecture, these include the attention query, key, value, and output matrices $\{W_{q,i}, W_{k,i}, W_{v,i}, W_{o,i}\}$ and the feed-forward network matrices $\{W_{gate,i}, W_{up,i}, W_{down,i}\}$.

To bridge the original residual dimension $D$ and the student's reduced dimension $D'$, we introduce a pair of learnable projection matrices for each weight matrix. Unlike LRC's random initialization, we employ the activation-aware initialization described in the main text. During training, the teacher's original weights remain frozen. The student effectively inherits the teacher's knowledge by performing the forward pass through these projection-wrapped modules. We then calculate the feature alignment loss $\mathcal{L}_{\text{REP}}$ based on the collected intermediate features, and obtain $\mathcal{L}_{\text{LM}}$ and $\mathcal{L}_{\text{LM}}$ using the final logits.

After the training is complete, the projection matrices are merged with the frozen weights to produce a hardware-friendly student model with a native residual dimension of $D'$.

# D. Theoretical Proof

## D.1. Proof of Lemma 4.1 [Evolution of Projection Matrices]

To ensure the rigor of the derivation, we first summarize the matrix calculus identities used in the proof, which are standard results in matrix analysis (Petersen et al., 2008).

**Lemma D.1** (Matrix Calculus Identities). *For any matrices $A, B, C$ with compatible dimensions, the following gradients hold:*

(i) $\frac{\partial \mathrm{Tr}(AB)}{\partial B} = A^\top$

(ii) $\frac{\partial \mathrm{Tr}(B^\top AB)}{\partial B} = (A + A^\top)B$

(iii) $\frac{\partial \mathrm{Tr}(CB^\top AB)}{\partial B} = A^\top BC^\top + ABC$

We now restate and prove the gradient flow dynamics for the projection matrices.

**Lemma D.2** (Evolution of Projection Matrices). *Under the reconstruction loss $\mathcal{L}$, the gradient flow dynamics for the projection matrices $O$ and $Q$ are governed by the following system of Ordinary Differential Equations (ODEs):*

$$\dot{O} = -\nabla_O \mathcal{L} = -Q^\top \Sigma(QO - I), \tag{14}$$
$$\dot{Q} = -\nabla_Q \mathcal{L} = -\Sigma(QO - I)O^\top. \tag{15}$$

*where $\dot{O}$ and $\dot{Q}$ denote the time-derivatives $\frac{dO}{dt}$ and $\frac{dQ}{dt}$ respectively, and $I$ is the identity matrix of the corresponding dimensions.*

*Proof.* First, we expand the reconstruction loss $\mathcal{L} = \frac{1}{2L}\|X - XQO\|_F^2$ using the property $\|A\|_F^2 = \mathrm{Tr}(A^\top A)$:

$$\begin{aligned}
\mathcal{L} &= \frac{1}{2L} \mathrm{Tr}\left((X - XQO)^\top(X - XQO)\right) \\
&= \frac{1}{2L} \mathrm{Tr}\left(X^\top X - X^\top XQO - O^\top Q^\top X^\top X + O^\top Q^\top X^\top XQO\right) \\
&= \frac{1}{2}\mathrm{Tr}(\Sigma) - \mathrm{Tr}(\Sigma QO) + \frac{1}{2}\mathrm{Tr}(O^\top Q^\top \Sigma QO),
\end{aligned} \tag{16}$$

where we use $\Sigma = \frac{1}{L}X^\top X$, the property $\mathrm{Tr}(A) = \mathrm{Tr}(A^\top)$ to combine the cross terms $\frac{1}{2L}(\mathrm{Tr}(X^\top XQO) + \mathrm{Tr}(O^\top Q^\top X^\top X)) = \mathrm{Tr}(\Sigma QO)$, and the cyclic property of the trace.

**Derivation of $\dot{O}$:** Applying the gradient operator $\nabla_O$ to (16):

$$\nabla_O \mathcal{L} = \nabla_O \left[-\mathrm{Tr}(\Sigma QO)\right] + \nabla_O \left[\frac{1}{2}\mathrm{Tr}(O^\top Q^\top \Sigma QO)\right].$$

Using Lemma D.1(i) with $A = \Sigma Q$ and $B = O$, the first term becomes $-(\Sigma Q)^\top = -Q^\top \Sigma$.
Using Lemma D.1(ii) with $B = O$ and $A = Q^\top \Sigma Q$, and noting that $A$ is symmetric since $\Sigma$ is symmetric, the second term becomes $\frac{1}{2}(A + A^\top)O = AO = Q^\top \Sigma QO$.
Combining these results:

$$\begin{aligned}
\nabla_O \mathcal{L} &= -Q^\top \Sigma + Q^\top \Sigma QO \\
&= Q^\top \Sigma(QO - I).
\end{aligned}$$

Thus, the flow is:

$$\dot{O} = -\nabla_O \mathcal{L} = -Q^\top \Sigma(QO - I). \tag{17}$$

**Derivation of $\dot{Q}$:** Applying the gradient operator $\nabla_Q$ to (16):

$$\nabla_Q \mathcal{L} = \nabla_Q \left[ -\text{Tr}(\Sigma QO) \right] + \nabla_Q \left[ \frac{1}{2} \text{Tr}(O^\top Q^\top \Sigma QO) \right].$$

For the first term, using the cyclic property $\text{Tr}(\Sigma QO) = \text{Tr}(O\Sigma Q)$, by Lemma D.1(i) with $A = O\Sigma$ and $B = Q$, we have $-\nabla_Q \text{Tr}(O\Sigma Q) = -(O\Sigma)^\top = -\Sigma O^\top$.

For the second term, we use the cyclic property $\text{Tr}(O^\top Q^\top \Sigma QO) = \text{Tr}(OO^\top Q^\top \Sigma Q)$. Applying Lemma D.1(iii) with $B = Q, A = \Sigma, C = OO^\top$:

$$\nabla_Q \left[ \frac{1}{2} \text{Tr}(CQ^\top AQ) \right] = \frac{1}{2}(A^\top QC^\top + AQC)$$
$$= \frac{1}{2}(\Sigma QOO^\top + \Sigma QOO^\top) = \Sigma QOO^\top,$$

where we use $\Sigma = \Sigma^\top$ and $(OO^\top) = (OO^\top)^\top$. Combining the terms:

$$\nabla_Q \mathcal{L} = -\Sigma O^\top + \Sigma QOO^\top$$
$$= \Sigma(QO - I)O^\top.$$

Thus, the gradient flow is:

$$\dot{Q} = -\nabla_Q \mathcal{L} = -\Sigma(QO - I)O^\top. \tag{18}$$

The proof is complete. $\square$

## D.2. Proof of Lemma D.4 [Balancedness Invariant]

To establish the balancedness invariant, we first state the fundamental matrix differentiation rules used in our derivation.

**Lemma D.3** (Matrix Differentiation Rules). *For any differentiable matrix-valued functions $A(t)$ and $B(t)$, the following identities hold:*

*(i) **Product Rule:** $\frac{d}{dt}(AB) = \dot{A}B + A\dot{B}$*

*(ii) **Transpose Rule:** $\frac{d}{dt}(A^\top) = (\dot{A})^\top$*

Now we provide the step-by-step proof for the conservation law:

**Lemma D.4** (Balancedness Invariant). *Under the reconstruction loss $\mathcal{L}$, for the gradient flow dynamics of $Q$ and $O$, the following conservation law holds for all $t \geq 0$:*

$$\frac{d}{dt}\left( Q(t)^\top Q(t) - O(t)O(t)^\top \right) = 0. \tag{19}$$

*Consequently, if the matrices are initialized such that $Q(0)^\top Q(0) = O(0)O(0)^\top$, they remain "balanced" throughout the entire optimization process, i.e., $Q(t)^\top Q(t) = O(t)O(t)^\top$ for all $t$.*

*Proof.* We aim to show that the time derivative of the difference $Q(t)^\top Q(t) - O(t)O(t)^\top$ is zero. Let $\Sigma = \frac{1}{L}X^\top X$ be the symmetric covariance matrix of $X$ ($\Sigma = \Sigma^\top$). From Lemma 4.1, the gradient flow dynamics are given by:

$$\dot{O} = -Q^\top \Sigma(QO - I), \tag{20}$$
$$\dot{Q} = -\Sigma(QO - I)O^\top. \tag{21}$$

**Step 1: Compute the derivative of $Q^\top Q$.** Applying the product rule (i) and transpose rule (ii) from Lemma D.3:

$$\frac{d}{dt}(Q^\top Q) = \dot{Q}^\top Q + Q^\top \dot{Q}. \tag{22}$$

Substituting the expression for $\dot{Q}$ from (21) and using the symmetry $\Sigma^\top = \Sigma$:

$$
\begin{aligned}
\dot{Q}^\top Q &= \left[-\Sigma(QO - I)O^\top\right]^\top Q \\
&= -O(QO - I)^\top \Sigma^\top Q \\
&= -O(QO - I)^\top \Sigma Q.
\end{aligned}
\tag{23}
$$

The second term is directly obtained by substituting $\dot{Q}$:

$$
Q^\top \dot{Q} = -Q^\top \Sigma(QO - I)O^\top.
\tag{24}
$$

Combining (23) and (24), we have:

$$
\frac{d}{dt}(Q^\top Q) = -O(QO - I)^\top \Sigma Q - Q^\top \Sigma(QO - I)O^\top.
\tag{25}
$$

**Step 2: Compute the derivative of $OO^\top$.** Similarly, for the product $OO^\top$, we have:

$$
\frac{d}{dt}(OO^\top) = \dot{O}O^\top + O\dot{O}^\top.
\tag{26}
$$

Substituting the expression for $\dot{O}$ from (20) into the first term:

$$
\dot{O}O^\top = -Q^\top \Sigma(QO - I)O^\top.
\tag{27}
$$

For the second term, we take the transpose of $\dot{O}$ and again use $\Sigma^\top = \Sigma$:

$$
\begin{aligned}
O\dot{O}^\top &= O\left[-Q^\top \Sigma(QO - I)\right]^\top \\
&= -O(QO - I)^\top \Sigma^\top Q \\
&= -O(QO - I)^\top \Sigma Q.
\end{aligned}
\tag{28}
$$

Combining 27 and 28, we obtain:

$$
\frac{d}{dt}(OO^\top) = -Q^\top \Sigma(QO - I)O^\top - O(QO - I)^\top \Sigma Q.
\tag{29}
$$

**Step 3: Conclusion.** Comparing (25) and (29), it is evident that:

$$
\frac{d}{dt}(Q^\top Q) = \frac{d}{dt}(OO^\top).
\tag{30}
$$

This implies that $\frac{d}{dt}(Q^\top Q - OO^\top) = 0$ for all $t \geq 0$. Integrating with respect to time, we find:

$$
Q(t)^\top Q(t) - O(t)O(t)^\top = Q(0)^\top Q(0) - O(0)O(0)^\top.
\tag{31}
$$

Therefore, if the matrices are initialized such that $Q(0)^\top Q(0) = O(0)O(0)^\top$, the equality $Q(t)^\top Q(t) = O(t)O(t)^\top$ is preserved throughout the entire optimization trajectory. $\square$

### D.3. Proof of Theorem 4.2 [Singular Value Dynamics]

We begin by stating two auxiliary lemmas that are essential for deriving the singular value dynamics.

**Lemma D.5** (Singular Value Evolution (Jing et al., 2021)). *Let $W(t)$ be a time-varying matrix and $W = U\Sigma V^\top$ be its Singular Value Decomposition. The time derivative of the $r$-th singular value $\sigma^r$ is given by:*

$$
\dot{\sigma}^r = (u^r)^\top \dot{W} v^r,
\tag{32}
$$

*where $u^r$ and $v^r$ are the $r$-th left and right singular vectors, respectively.*

**Lemma D.6** (Balance Property and SVD Identities). *Given the product matrix $M = QO$ with SVD $M = U_M E_M V_M^\top$ and assuming the balanced condition $Q^\top Q = OO^\top$, the symmetric components satisfy:*

$$O^\top O = (M^\top M)^{1/2} = V_M E_M V_M^\top, \tag{33}$$

$$QQ^\top = (MM^\top)^{1/2} = U_M E_M U_M^\top. \tag{34}$$

*Furthermore, for any $r$-th singular vector pair $(u_M^r, v_M^r)$, the following fundamental SVD identities hold:*

(i) **Vector Mapping:** $Mv_M^r = \sigma_M^r u_M^r$ and $(u_M^r)^\top M = \sigma_M^r (v_M^r)^\top.$

(ii) **Spectral Projection:** $V_M E_M V_M^\top v_M^r = \sigma_M^r v_M^r$ and $(u_M^r)^\top U_M E_M U_M^\top = \sigma_M^r (u_M^r)^\top.$

Now we provide the step-by-step proof for the following Theorem:

**Theorem D.7** (Singular Value Dynamics). *Given the balanced condition $Q^\top Q = OO^\top$, the evolution of the $r$-th singular value $\sigma_M^r$ of the product matrix $M = QO$ follows:*

$$\dot{\sigma}_M^r = 2\sigma_M^r (u_M^r)^\top \Sigma (v_M^r - \sigma_M^r u_M^r). \tag{35}$$

*where $u_M^r$ and $v_M^r$ are the $r$-th left and right singular vectors of $M$, respectively.*

*Proof.* We aim to derive the dynamics of $\sigma_M^r$. First, we compute the time derivative of $M = QO$. By applying the product rule and substituting the gradient flow dynamics:

$$\begin{aligned} \dot{M} &= \dot{Q}O + Q\dot{O} \\ &= \left[ -\Sigma(QO - I)O^\top \right] O + Q \left[ -Q^\top \Sigma(QO - I) \right] \\ &= -\Sigma(M - I)(O^\top O) - (QQ^\top)\Sigma(M - I). \end{aligned} \tag{36}$$

Using the identities (33) and (34) from Lemma D.6, we substitute the $O^\top O$ and $QQ^\top$ with their SVD formulations:

$$\dot{M} = -\Sigma(M - I)V_M E_M V_M^\top - U_M E_M U_M^\top \Sigma(M - I). \tag{37}$$

According to Lemma D.5, the evolution of $\sigma_M^r$ is obtained by projecting $\dot{M}$ onto the left and right singular vectors:

$$\begin{aligned} \dot{\sigma}_M^r &= (u_M^r)^\top \dot{M} v_M^r \\ &= (u_M^r)^\top \left[ -\Sigma(M - I)V_M E_M V_M^\top - U_M E_M U_M^\top \Sigma(M - I) \right] v_M^r \\ &= -(u_M^r)^\top \Sigma(M - I) \underbrace{(V_M E_M V_M^\top v_M^r)}_{\text{TermA}} - \underbrace{((u_M^r)^\top U_M E_M U_M^\top)}_{\text{TermB}} \Sigma(M - I)v_M^r. \end{aligned} \tag{38}$$

By applying the Spectral Projection property (ii) from Lemma D.6, where Term A simplifies to $\sigma_M^r v_M^r$ and Term B simplifies to $\sigma_M^r (u_M^r)^\top$, we have:

$$\begin{aligned} \dot{\sigma}_M^r &= -(u_M^r)^\top \Sigma(M - I)(\sigma_M^r v_M^r) - (\sigma_M^r u_M^r)^\top \Sigma(M - I)v_M^r \\ &= -\sigma_M^r \left[ (u_M^r)^\top \Sigma(M - I)v_M^r \right] - \sigma_M^r \left[ (u_M^r)^\top \Sigma(M - I)v_M^r \right] \\ &= -2\sigma_M^r \left[ (u_M^r)^\top \Sigma(M - I)v_M^r \right]. \end{aligned} \tag{39}$$

To further simplify the bracketed term, we expand $(M - I)$:

$$(u_M^r)^\top \Sigma(M - I)v_M^r = (u_M^r)^\top \Sigma M v_M^r - (u_M^r)^\top \Sigma v_M^r. \tag{40}$$

Applying the Vector Mapping property (i) from Lemma D.6 ($Mv_M^r = \sigma_M^r u_M^r$), we obtain:

$$\begin{aligned} (u_M^r)^\top \Sigma(M - I)v_M^r &= (u_M^r)^\top \Sigma(\sigma_M^r u_M^r) - (u_M^r)^\top \Sigma v_M^r \\ &= \sigma_M^r (u_M^r)^\top \Sigma u_M^r - (u_M^r)^\top \Sigma v_M^r. \end{aligned} \tag{41}$$

Substituting this expansion back into equation (39):

$$\dot{\sigma}_M^r = -2\sigma_M^r \left[ \sigma_M^r (u_M^r)^\top \Sigma u_M^r - (u_M^r)^\top \Sigma v_M^r \right]$$

$$= 2\sigma_M^r \left[ (u_M^r)^\top \Sigma v_M^r - \sigma_M^r (u_M^r)^\top \Sigma u_M^r \right] \tag{42}$$

$$= 2\sigma_M^r (u_M^r)^\top \Sigma \left( v_M^r - \sigma_M^r u_M^r \right). \tag{43}$$

This establishes the specific dynamics of the singular values under the balanced gradient flow. $\qquad\square$

### D.4. Proof of Lemma D.10 [Spectral Coupling]

To prove the spectral coupling property under the balancedness condition, we first state two fundamental results from matrix analysis that characterize the relationship between singular values and eigenvalues of matrix products.

**Lemma D.8** (Singular Value-Eigenvalue Relation (Horn & Johnson, 2012)). *For any matrix $A \in \mathbb{R}^{m \times n}$, let $\sigma_i(A)$ denote its $r$-th largest singular value. Then:*

$$\sigma_r(A) = \sqrt{\lambda_r(A^\top A)} = \sqrt{\lambda_r(AA^\top)}, \tag{44}$$

*where $\lambda_r(\cdot)$ denotes the $i$-th largest eigenvalue of a symmetric positive semi-definite matrix.*

**Lemma D.9** (Commutation of Eigenvalues (Horn & Johnson, 2012)). *For any two matrices $B \in \mathbb{R}^{m \times n}$ and $C \in \mathbb{R}^{n \times m}$, the non-zero eigenvalues of the product $BC$ are identical to the non-zero eigenvalues of the product $CB$. That is:*

$$\lambda_r(BC) = \lambda_r(CB), \quad \forall \lambda_r \neq 0. \tag{45}$$

We now provide the formal proof of the following Lemma:

**Lemma D.10** (Spectral Coupling). *Under the balancedness condition $Q^\top Q = OO^\top$, let the singular values of $Q, O$, and $M = QO$ be $\{\sigma_Q^r\}$, $\{\sigma_O^r\}$, and $\{\sigma_M^r\}$ respectively. It holds that:*

$$\sigma_Q^r = \sigma_O^r = \sqrt{\sigma_M^r}, \quad \text{for} r = 1, \ldots, D'. \tag{46}$$

*Proof.* Recall the balancedness condition $Q^\top Q = OO^\top$, where $Q \in \mathbb{R}^{D \times D'}$ and $O \in \mathbb{R}^{D' \times D}$.

**Step 1: Establishing $\sigma_Q^r = \sigma_O^r$.** By applying Lemma D.8 to the matrices $Q$ and $O$, the squares of their singular values are given by the eigenvalues of $Q^\top Q \in \mathbb{R}^{D' \times D'}$ and $OO^\top \in \mathbb{R}^{D' \times D'}$, respectively:

$$(\sigma_Q^r)^2 = \lambda_r(Q^\top Q), \tag{47}$$

$$(\sigma_O^r)^2 = \lambda_r(OO^\top). \tag{48}$$

Substituting the balancedness condition $Q^\top Q = OO^\top$ into the above, we have $\lambda_r(Q^\top Q) = \lambda_r(OO^\top)$. Since singular values are by definition non-negative, it immediately follows that:

$$\sigma_Q^r = \sigma_O^r, \quad \text{for} \quad r = 1, \ldots, D'. \tag{49}$$

**Step 2: Relating $\sigma_M^r$ to the balanced components.** Now consider the product matrix $M = QO \in \mathbb{R}^{D \times D}$. By Lemma D.8, the squared singular values $(\sigma_M^r)^2$ are the eigenvalues of $M^\top M$:

$$(\sigma_M^r)^2 = \lambda_r(M^\top M) = \lambda_r \left( (QO)^\top (QO) \right) = \lambda_r(O^\top Q^\top QO). \tag{50}$$

Substituting the balancedness condition $Q^\top Q = OO^\top$ into the expression above:

$$(\sigma_M^r)^2 = \lambda_r(O^\top (OO^\top) O) = \lambda_r(O^\top OO^\top O) = \lambda_r \left( (O^\top O)^2 \right). \tag{51}$$

For any symmetric matrix, the eigenvalues of its square are the squares of its eigenvalues. Thus:

$$(\sigma_M^r)^2 = \left( \lambda_r(O^\top O) \right)^2. \tag{52}$$

By invoking Lemma D.9, the non-zero eigenvalues of $O^\top O \in \mathbb{R}^{D \times D}$ are identical to those of $OO^\top \in \mathbb{R}^{D' \times D'}$. Therefore, we have $\lambda_r(O^\top O) = \lambda_r(OO^\top)$. Substituting this into Eq. (52) yields:

$$(\sigma_M^r)^2 = \left(\lambda_r(OO^\top)\right)^2. \tag{53}$$

Taking the square root of both sides (noting that $\sigma_M^r \geq 0$ and $\lambda_r(OO^\top) \geq 0$ as $OO^\top$ is positive semi-definite):

$$\sigma_M^r = \lambda_r(OO^\top). \tag{54}$$

**Conclusion.** Finally, we link the singular values of $M$ to those of $Q$ and $O$. From Step 1, we know $(\sigma_O^r)^2 = \lambda_r(OO^\top)$ and $(\sigma_Q^r)^2 = \lambda_r(Q^\top Q)$. Combining this with Eq. (54), we obtain:

$$\sigma_M^r = (\sigma_O^r)^2 = (\sigma_Q^r)^2, \tag{55}$$

which is equivalent to $\sigma_Q^r = \sigma_O^r = \sqrt{\sigma_M^r}$ for all $r = 1, \ldots, D'$. This completes the proof. $\qquad \square$

### D.5. Proof of Theorem D.13 [Representation Distinguishability Bound]

Assume the representation matrix $X \in \mathbb{R}^{L \times D}$ is row-normalized and zero-centered. Recall that $\Sigma = \frac{1}{L} X^\top X$ denote the covariance matrix and $\{\lambda_j\}_{j=1}^D$ are the eigenvalues of $X^\top X$. The eRank of representations $X$ is defined as:

$$\mathrm{eRank}(X) = \exp\left(-\sum_{j=1}^D p_j \log p_j\right), \tag{56}$$

where $p_j = \frac{\lambda_j}{\sum_{k=1}^D \lambda_k}$.

To prove the lower bound of the maximum absolute cosine similarity, we first establish two lemmas based on standard inequalities and matrix properties.

**Lemma D.11** (Lower Bound of Collision Probability via Jensen's Inequality). *For the probability distribution $\{p_j\}_{j=1}^D$ defined in Eq. 56, where $\sum_{j=1}^D p_j = 1$, the following inequality holds:*

$$\sum_{j=1}^D p_j^2 \geq \frac{1}{\mathrm{eRank}(X)}. \tag{57}$$

*Proof.* Consider the convex function $\phi(a) = -\log a$. By Jensen's Inequality (McShane, 1937), we have $\phi(\mathbb{E}[Y]) \leq \mathbb{E}[\phi(Y)]$. Let $Y$ be a random variable that takes the value $p_j$ with probability $p_j$. Then:

$$-\log\left(\sum_{j=1}^D p_j^2\right) \leq \sum_{j=1}^D p_j(-\log p_j) = \log(\mathrm{eRank}(X)). \tag{58}$$

Taking the exponential and inverting terms, we obtain $\sum_{j=1}^D p_j^2 \geq \frac{1}{\mathrm{eRank}(X)}$. $\qquad \square$

**Lemma D.12** (Spectral Properties of $X$). *(Petersen et al., 2008; Horn & Johnson, 2012) For a row-normalized matrix $X \in \mathbb{R}^{L \times D}$ where $\|X^l\|_2 = 1$, the Gram matrix $XX^\top \in \mathbb{R}^{L \times L}$ satisfies:*

$$\mathrm{Tr}(XX^\top) = \sum_{l=1}^L \|X^l\|_2^2 = L, \quad \text{and} \quad \|XX^\top\|_F^2 = \sum_{j=1}^D \lambda_j^2, \tag{59}$$

*where $\{\lambda_j\}_{j=1}^D$ are the eigenvalues of $X^\top X$.*

Now we begin to prove the following Theorem:

**Theorem D.13** (Representation Distinguishability Bound). *Assume the representation matrix $X \in \mathbb{R}^{L \times D}$ is row-normalized and zero-centered. Let $\rho_{l_1, l_2} = |X^{l_1} X^{l_2\top}|$ denote the absolute cosine similarity between features of $l_1$-th and $l_2$-th tokens. It holds that:*

$$\max_{l_1 \neq l_2} \rho^{l1, l2} \geq \sqrt{\frac{1}{L-1} \left( \frac{L}{\mathrm{eRank}(X)} - 1 \right)}. \tag{60}$$

*Proof.* Based on the definition, $\rho_{l_1, l_2} = |X^{l_1} X^{l_2\top}|$. The squared Frobenius norm of $XX^\top$ can be decomposed into diagonal and off-diagonal terms:

$$\|XX^\top\|_F^2 = \sum_{l=1}^{L} (X^l X^{l\top})^2 + \sum_{l_1 \neq l_2} (X^{l_1} X^{l_2\top})^2. \tag{61}$$

Since $X$ is row-normalized, $X^l X^{l\top} = 1$, thus the diagonal sum is $L$. It follows that:

$$\sum_{l_1 \neq l_2} \rho_{l_1, l_2}^2 = \|XX^\top\|_F^2 - L. \tag{62}$$

The number of off-diagonal terms in the $L \times L$ matrix is $L^2 - L$. Since the maximum value in a set is at least as large as the average value, we have:

$$\max_{l_1 \neq l_2} \rho_{l_1, l_2}^2 \geq \frac{1}{L^2 - L} \sum_{l_1 \neq l_2} \rho_{l_1, l_2}^2 = \frac{1}{L^2 - L} \left( \|XX^\top\|_F^2 - L \right). \tag{63}$$

Next, we relate $\|XX^\top\|_F^2$ to eRank. From Lemma D.12, $\|XX^\top\|_F^2 = \sum_{j=1}^{D} \lambda_j^2$. Noting that $\sum_{k=1}^{D} \lambda_k = \mathrm{Tr}(XX^\top) = L$ and $p_j = \frac{\lambda_j}{\sum_{k=1}^{D} \lambda_k}$, we have $\lambda_j = L p_j$. Thus:

$$\|XX^\top\|_F^2 = \sum_{j=1}^{D} (L p_j)^2 = L^2 \sum_{j=1}^{D} p_j^2. \tag{64}$$

Applying Lemma D.11, $\sum_{j=1}^{D} p_j^2 \geq \frac{1}{\mathrm{eRank}(X)}$, so $\|XX^\top\|_F^2 \geq \frac{L^2}{\mathrm{eRank}(X)}$. Substituting this into Eq. 63:

$$\max_{l_1 \neq l_2} \rho_{l_1, l_2}^2 \geq \frac{1}{L^2 - L} \left( \frac{L^2}{\mathrm{eRank}(X)} - L \right) \tag{65}$$

$$= \frac{L}{L(L-1)} \left( \frac{L}{\mathrm{eRank}(X)} - 1 \right) \tag{66}$$

$$= \frac{1}{L-1} \left( \frac{L}{\mathrm{eRank}(X)} - 1 \right). \tag{67}$$

Taking the square root of both sides, we reach the final conclusion:

$$\max_{l_1 \neq l_2} \rho_{l_1, l_2} \geq \sqrt{\frac{1}{L-1} \left( \frac{L}{\mathrm{eRank}(X)} - 1 \right)}. \tag{68}$$

$\square$

## D.6. Proof of Theorem 4.3 [Probability Distinguishability Bound]

**Lemma D.14** (Norm Inequalities and Softmax Lipschitzness). *(Horn & Johnson, 2012; Gao & Pavel, 2017) Let $\mathbf{z}_1, \mathbf{z}_2 \in \mathbb{R}^n$ be vectors and $A \in \mathbb{R}^{n \times m}$ be a matrix. The following properties hold:*

  *(i) **Norm Equivalence:** For any vector $a \in \mathbb{R}^n$, the $L_1$ and $L_2$ norms satisfy $\|a\|_1 \leq \sqrt{n}\|a\|_2$.*

  *(ii) **Matrix Lipschitzness:** For the vector $L_2$ norm, $\|aA\|_2 \leq \|a\|_2\|A\|_2$, where $\|A\|_2$ is the **spectral norm** (the largest singular value of A).*

  *(iii) **Softmax Lipschitzness:** The $\mathrm{Softmax}$ function, defined as $\mathrm{Softmax}(\mathbf{z})_i = \frac{e^{z_i}}{\sum_j e^{z_j}}$, is 1-Lipschitz continuous with respect to the $L_2$ norm, i.e., $\|\mathrm{Softmax}(\mathbf{z}_1) - \mathrm{Softmax}(\mathbf{z}_2)\|_2 \leq \|\mathbf{z}_1 - \mathbf{z}_2\|_2$.*

  *(iv) **Cosine Distance Identity:** For any two unit vectors $a, b$ (where $\|a\|_2 = \|b\|_2 = 1$) that are **co-directional** (i.e., $\langle a, b \rangle \geq 0$), their Euclidean distance and absolute cosine similarity $\rho = |\langle a, b \rangle|$ are related by $\|a - b\|_2 = \sqrt{2 - 2\rho}$.*

We then prove the following bound:

**Theorem D.15** (Probability Distinguishability Bound). *Let $X^{l_1}, X^{l_2} \in \mathbb{R}^D$ be the features of the $l_1$-th and $l_2$-th tokens from a row-normalized and zero-centered representation matrix $X \in \mathbb{R}^{L \times D}$. $P^l \in \mathbb{R}^{\mathrm{Voc}}$ be the output probability distribution defined as $P^l = \mathrm{Softmax}(Z^l)$, where $Z^l = \mathrm{RMSNorm}_{\mathrm{final}}(X^l)W_u$ are the logits, $W_u \in \mathbb{R}^{D \times \mathrm{Voc}}$ is the unembedding weight, and $g_{\mathrm{final}}$ is the learnable scaling parameter of $\mathrm{RMSNorm}_{\mathrm{final}}$. The minimum TV distance between any two different token predictions is upper-bounded as:*

$$\min_{l_1 \neq l_2} d_{TV}(P^{l_1}, P^{l_2}) \leq \frac{1}{2}\sqrt{\mathrm{Voc}}\|\mathrm{diag}(g_{\mathrm{final}})W_u\|_2$$

$$\cdot \sqrt{2 - 2\sqrt{\frac{1}{L-1}\left(\frac{L}{\mathrm{eRank}(X)} - 1\right)}}, \tag{69}$$

*where $\|\mathrm{diag}(g_{\mathrm{final}})W_u\|_2$ denotes the spectral norm of the scaled unembedding weight matrix.*

*Proof.* By the definition of Total Variation (TV) distance for discrete distributions:

$$d_{TV}(P^{l_1}, P^{l_2}) = \frac{1}{2}\sum_{j=1}^{\mathrm{Voc}} |P_j^{l_1} - P_j^{l_2}| = \frac{1}{2}\|P^{l_1} - P^{l_2}\|_1. \tag{70}$$

Using the norm equivalence property (Lemma D.14 (i)):

$$d_{TV}(P^{l_1}, P^{l_2}) \leq \frac{1}{2}\sqrt{\mathrm{Voc}}\|P^{l_1} - P^{l_2}\|_2. \tag{71}$$

Substituting $P^l = \mathrm{Softmax}(Z^l)$ and applying the Softmax Lipschitzness property (Lemma D.14 (iii)), we can upper-bound the $L_2$ distance between distributions by the $L_2$ distance between their corresponding logits:

$$\|P^{l_1} - P^{l_2}\|_2 = \|\mathrm{Softmax}(Z^{l_1}) - \mathrm{Softmax}(Z^{l_2})\|_2 \leq \|Z^{l_1} - Z^{l_2}\|_2. \tag{72}$$

In modern Transformer architectures, the logits are typically computed as $Z^l = \mathrm{RMSNorm}_{\mathrm{final}}(X^l)W_u$. Since $X^l$ is already row-normalized, the normalization layer simplifies to an element-wise scaling by $g_{\mathrm{final}}$, written as $Z^l = X^l \mathrm{diag}(g_{\mathrm{final}})W_u$. Applying the Matrix Lipschitzness property (Lemma D.14 (ii)):

$$\|Z^{l_1} - Z^{l_2}\|_2 = \|(X^{l_1} - X^{l_2})\mathrm{diag}(g_{\mathrm{final}})W_u\|_2$$
$$\leq \|X^{l_1} - X^{l_2}\|_2 \cdot \|\mathrm{diag}(g)W_u\|_2. \tag{73}$$

Combining Eq. (71), (72), and (73), we have:

$$d_{TV}(P^{l_1}, P^{l_2}) \leq \frac{1}{2}\sqrt{\mathrm{Voc}}\|\mathrm{diag}(g_{\mathrm{final}})W_u\|_2 \cdot \|X^{l_1} - X^{l_2}\|_2. \tag{74}$$

To find the bound for the *minimum* TV distance, we consider the most similar pair of tokens. Assuming they are co-directional due to the empirical "cone effect" in LLMs, we apply Lemma D.14 (iv):

$$\|X^{l_1} - X^{l_2}\|_2 = \sqrt{2 - 2\rho_{l_1, l_2}}, \tag{75}$$

where $\rho_{l_1, l_2} = |X^{l_1} X^{l_2 \top}|$ is the absolute cosine similarity. Then:

$$\min_{l_1 \neq l_2} d_{TV}(P^{l_1}, P^{l_2}) \leq \frac{1}{2} \sqrt{\mathrm{Voc}} \|\mathrm{diag}(g_{\mathrm{final}}) W_u\|_2 \cdot \sqrt{2 - 2 \max_{l_1 \neq l_2} \rho_{l_1, l_2}}. \tag{76}$$

Finally, invoking Theorem D.13 to bound the maximum cosine similarity via $\mathrm{eRank}(X)$:

$$\max_{l_1 \neq l_2} \rho_{l_1, l_2} \geq \sqrt{\frac{1}{L-1}\left(\frac{L}{\mathrm{eRank}(X)} - 1\right)}. \tag{77}$$

Substituting this into the inequality, we obtain the final bound:

$$\min_{l_1 \neq l_2} d_{TV}(P^{l_1}, P^{l_2}) \leq \frac{1}{2} \sqrt{\mathrm{Voc}} \|\mathrm{diag}(g_{\mathrm{final}}) W_u\|_2$$
$$\cdot \sqrt{2 - 2\sqrt{\frac{1}{L-1}\left(\frac{L}{\mathrm{eRank}(X)} - 1\right)}}. \tag{78}$$

This completes the proof. $\qquad\square$

### D.7. Proof of Theorem 4.4 [Binary State Collapse]

We provide a formal proof of the Binary State Collapse by analyzing the geometric constraints on the rows of a rank-1 matrix.

**Lemma D.16** (Properties of Shannon Entropy (Q. et al., 2006)). *For a probability distribution $p = (p_1, \ldots, p_D)$, the Shannon entropy $H(p) = -\sum_{j=1}^{D} p_j \log p_j$ is a measure of uncertainty that satisfies: $H(p) = 0$ if and only if $p$ is a one-hot distribution, where one element $p_k = 1$ and all other $p_j = 0$ for $j \neq k$.*

**Theorem D.17** (Binary State Collapse). *Assume the representation matrix $X \in \mathbb{R}^{L \times D}$ is zero-centered and row-normalized. As $\mathrm{eRank}(X) \to 1$, the rank of $X$ becomes 1. For any $l$-th token, its representation $X^l$ satisfies $X^l = X^1$ or $X^l = -X^1$. Consequently, the corresponding logits satisfy $Z^l = Z^1$ or $Z^l = \mathrm{Norm}_{\mathrm{final}}(-X^1)W_u$.*

*Proof.* **Step 1: Spectral Convergence to Rank-1.** From Definition 4, $\mathrm{eRank}(X) = \exp(H(p))$, where $p$ is the normalized eigenvalue distribution of the covariance matrix $\Sigma = \frac{1}{L} X^\top X$. As $\mathrm{eRank}(X) \to 1$, we have $H(p) \to 0$. By **Lemma D.16**, this means only the first eigenvalue $\lambda_1$ is non-zero ($\lambda_1 > 0$ and $\lambda_j = 0$ for $j > 1$). Thus, $\mathrm{rank}(\Sigma) = 1$. Since $\mathrm{rank}(X) = \mathrm{rank}(X^\top X) = \mathrm{rank}(\Sigma)$ (Petersen et al., 2008), it follows that $\mathrm{rank}(X) = 1$.

**Step 2: Binary Geometric Collapse.** In any matrix $X$ where $\mathrm{rank}(X) = 1$, all non-zero rows must be linearly dependent. This means any $l$-th row $X^l$ can be expressed as a scalar multiple of the first row $X^1$:

$$X^l = c_l X^1, \quad \mathrm{for\ some\ scalar\ } c_l \in \mathbb{R}. \tag{79}$$

By the row-normalization assumption, both $X^l$ and $X^1$ must have a unit norm ($\|X^l\|_2 = 1$ and $\|X^1\|_2 = 1$). Substituting the scalar relationship into the norm equation gives:

$$\|c_l X^1\|_2 = |c_l| \cdot \|X^1\|_2 = |c_l| \cdot 1 = 1. \tag{80}$$

This directly implies $|c_l| = 1$, which means $c_l \in \{1, -1\}$. Therefore, every token representation $X^l$ in the sequence must satisfy:

$$X^l = X^1 \quad \text{or} \quad X^l = -X^1. \tag{81}$$

**Step 3: Binarization of Output Logits.** The logits for each token are computed through the unembedding layer $Z^l = \mathrm{Norm}_{\mathrm{final}}(X^l)W_u$ (Eq. 2). Given the binary states of $X^l$ derived in Step 2:

1. If $X^l = X^1$, then $Z^l = \text{Norm}_{\text{final}}(X^1)W_{\text{u}} = Z^1$.

2. If $X^l = -X^1$, then $Z^l = \text{Norm}_{\text{final}}(-X^1)W_{\text{u}}$.

Thus, the sequence of logits $Z$ collapses into at most two distinct vectors across all $L$ positions. This binary restriction mitigates the model from assigning unique, context-dependent probabilities to tokens, leading to the loss of reasoning ability. $\square$

### D.8. Proofs of Theorem 5.1 [Vanishing Initial Dynamics]

We begin by establishing the algebraic properties of the selection matrix $H$ and the resulting product matrix $M(0) = Q(0)O(0) = HH'$ at initialization.

**Lemma D.18** (Properties of Selection Matrices). *Let $H \in \mathbb{R}^{D \times D'}$ be a selection matrix defined by an index set $G = \{G_1, \ldots, G_{D'}\} \subset \{1, \ldots, D\}$, where $H_{ij} = 1$ if $i = G_j$. Then:*

*(i) $H^\top H = I_{D'}$, where $I_{D'}$ is the $D' \times D'$ identity matrix.*

*(ii) $M(0) = HH^\top \in \mathbb{R}^{D \times D}$ is a symmetric projection matrix.*

*(iii) The singular values $\sigma^r_{HH^\top}$ of $HH^\top$ satisfy $\sigma^r_{HH^\top} \in \{0, 1\}$ for all $r \in \{1, \ldots, D\}$. Specifically, there are $D'$ singular values equal to $1$ and $D - D'$ singular values equal to $0$.*

*Proof.* (i) Since each column of $H$ is a distinct standard basis vector $e_{G_j}$, the columns are orthonormal, hence $H^\top H = I$. (ii) Symmetry follows from $(HH^\top)^\top = (H^\top)^\top H^\top = HH^\top$. (iii) For a symmetric projection matrix, since the projection property follows from $(HH^\top)^2 = H(H^\top H)H^\top = HIH^\top = HH^\top$, the eigenvalues are restricted to $\{0, 1\}$ (Petersen et al., 2008). Given $HH^\top$ is symmetric positive semi-definite (PSD), its singular values are identical to its eigenvalues, thus $\sigma^r_M \in \{0, 1\}$. $\square$

**Lemma D.19** (SVD of Symmetric PSD Matrices (Horn & Johnson, 2012)). *For any symmetric PSD matrix $A$, its singular value decomposition (SVD) $A = UEV^\top$ can be chosen such that $U = V$. In this case, for any singular value $\sigma^r$, the corresponding left and right singular vectors satisfy $u^r = v^r$.*

**Theorem D.20** (Vanishing Initial Dynamics). *Under our activation-preserved initialization, at $t = 0$, the singular values of $M$ satisfy $\sigma^r_M \in \{0, 1\}$. Furthermore, their initial derivatives vanish, i.e., $\dot{\sigma}^r_M = 0$ for all $r \in \{1, \ldots, D\}$.*

*Proof.* At initialization $t = 0$, the product matrix is given by $M(0) = Q(0)O(0) = HH^\top$. By Lemma D.18 (iii), it follows immediately that:

$$\sigma^r_M(0) \in \{0, 1\}, \quad \forall r \in \{1, \ldots, D\}. \tag{82}$$

This proves the first part of the theorem.

To prove $\dot{\sigma}^r_M = 0$, we substitute the initial state into the singular value dynamics from Theorem 4.2:

$$\dot{\sigma}^r_M = 2\sigma^r_M (u^r_M)^\top \Sigma (v^r_M - \sigma^r_M u^r_M). \tag{83}$$

We analyze the dynamics based on the two possible values of $\sigma^r_M(0)$:

**Case 1:** $\sigma^r_M(0) = 0$. The leading factor $\sigma^r_M$ in Equation (83) is zero, which directly yields $\dot{\sigma}^r_M = 0$.

**Case 2:** $\sigma^r_M(0) = 1$. From Lemma D.18 (ii), $M(0)$ is a symmetric PSD matrix. Applying Lemma D.19, the singular vectors satisfy $u^r_M = v^r_M$. Substituting $\sigma^r_M = 1$ and $v^r_M = u^r_M$ into Equation (83):

$$\dot{\sigma}^r_M = 2(1) \cdot (u^r_M)^\top \Sigma (u^r_M - (1)u^r_M) \tag{84}$$

$$= 2(u^r_M)^\top \Sigma (0) \tag{85}$$

$$= 0. \tag{86}$$

Combining both cases, we conclude $\dot{\sigma}^r_M = 0$ for all $r$ at $t = 0$, completing the proof. $\square$

# E. Representation Geometry of Depth-reduced EDistilled LLMs

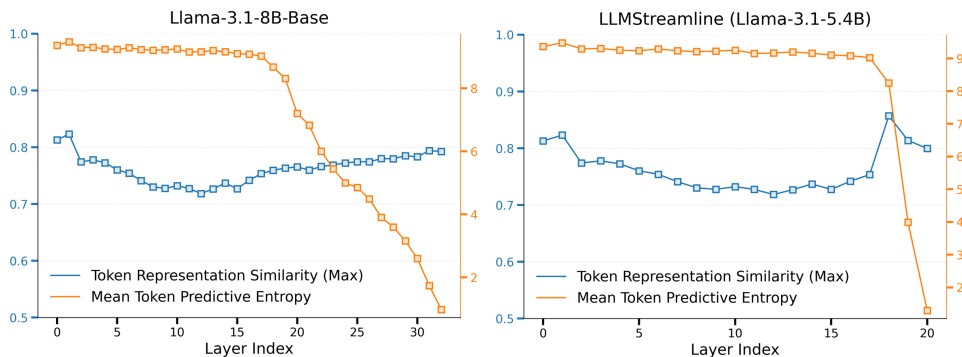

*Figure 5.* **Comparative analysis of representation geometry and predictive confidence** between the teacher (Llama-3.1-8B-Base) and the depth-reduced student (5.4B). **Top Row:** Layer-wise sequence-averaged maximum absolute cosine similarity (excluding self), indicating representation separability. **Bottom Row:** Sequence-averaged per-token entropy. While the teacher exhibits a gradual transition, the student manifests a sharp *entropy cliff* synchronized with a *similarity spike*, signifying a structural collapse of the reasoning trajectory.

We further analyze the co-evolution of the maximum token representation similarity and predictive uncertainty (Entropy defined in Appendix I.2) across layers. As illustrated in Figure 5, there is a notable divergence in how the teacher and depth-reduced student process information.

**Entropy Cliff.** In the teacher model (Llama-3.1-8B-Base), the token entropy (bottom-left) exhibits a smooth, multi-stage decay, suggesting a gradual refinement process where the model progressively resolves ambiguity (Gupta et al., 2025; Tan et al., 2025). In contrast, the depth-reduced student (bottom-right) maintains a high-entropy "exploration" state for over 80% of its depth, followed by a precipitous *entropy cliff* in the final three layers. This indicates that the model is forced to compress the entire transition from exploration to commitment into a dangerously narrow computational window. Such shallow Transformer architectures are known to favor shortcut-style solutions that defer decisive computation rather than executing stepwise reasoning (Liu et al., 2022), and tend to rely on strong statistical co-occurrence relationships rather than combining input prompts for combinatorial reasoning (**?**).

**Contextual Confusion.** Crucially, this entropy drop in the student is perfectly synchronized with a sharp spike in similarity in the representation space (top-right). While the teacher's token similarity increases moderately and smoothly, the student's representations become markedly less separable precisely at the point of decision-making. This confusion often makes LLMs prone to hallucinations during reasoning (**?**Fu et al., 2026).

# F. Experiment Settings

## F.1. Training Hyperparameter Settings

In Table 8, we present the hyperparameters we employ. Except for the temperature coefficient in the logits loss being set to 1, the rest remain entirely consistent with LRC.

## F.2. Datasets

Following the methodology of LRC, we curate three distinct data mixtures for our training process: Mixed-1.1 (10B) for RED-1.5B-Base, Mixed-1.1 (20B) for RED-1.7B-Base, and Mixed-2.0 (18B) for RED-4B-Base. The detailed composition and token counts are summarized in Table 9. All datasets utilized are open-sourced. It is worth noting that the training process does not necessarily exhaust the entire shuffled dataset. For instance, RED-4B-Base is trained on 18B tokens, even though the Mixed-2.0 pool contains a total of 21.5B tokens.

For calibration and internal representation analysis (including token entropy, representation eRank, similarity, and logits), we utilize samples from three subsets of the Nemotron-Pretraining-Dataset: 'Nemotron-CC-Diverse-QA', 'Nemotron-SFT-Code', and 'Nemotron-SFT-General'. We select 5 samples per subset for calibration and 1 sample per subset for representation analysis. Furthermore, 5 samples from SlimOrca are extracted for the linear autoencoder experiments. The full list of Hugging Face ID for these datasets is provided in Table 10.

*Table 8.* Training hyperparameters for our experiments.

| Model | RED-1.5B-Base | RED-1.7B-Base | RED-4B-Base |
|---|---|---|---|
| Teacher | Llama-3.2-3B-Instruct | Qwen2.5-3B-Instruct | Qwen2.5-7B-Instruct |
| Trained Tokens | 10B | 20B | 18B |
| Pre-train Dataset | Mixed 1.1 | Mixed 1.1 | Mixed 2.0 |
| Pre-trained Tokens | 10B | 20B | 18B |
| Teacher Hidden Size | 3,072 | 2,048 | 3,584 |
| Student Hidden Size | 1,536 | 1,200 | 2,048 |
| Sequence Length | 2,048 | 2,048 | 2,048 |
| Batch Size (tokens) | 49,152 | 32,768 | 32,768 |
| Clone Loss Weight ($\alpha$) | 0.2 | 0.5 | 0.5 |
| Learning Rate (Pre-train) | $1.0 \times 10^{-4}$ | $6.7 \times 10^{-5}$ | $1.0 \times 10^{-4}$ |
| LR Scheduler | Linear | Linear | Linear |
| Warm-up Ratio | 0.005 | 0.005 | 0.005 |
| Optimizer | Adam | Adam | Adam |
| Adam $\beta_1$ | 0.9 | 0.9 | 0.9 |
| Adam $\beta_2$ | 0.999 | 0.999 | 0.999 |
| Temperature for $\mathcal{L}_{\mathrm{KL}}$ | 1 | 1 | 1 |
| RMSNorm $\epsilon$ | $1.0 \times 10^{-5}$ | $1.0 \times 10^{-5}$ | $1.0 \times 10^{-5}$ |
| GPUs | $8 \times$ H800 | $8 \times$ H800 | $8 \times$ H800 |
| Training Time | 30 Hours | 79 Hours | 140 Hours |

*Table 9.* Training dataset composition (number of exact training tokens).

| Training Dataset | Mixed-1.1 (10B) | Mixed-1.1 (20B) | Mixed-2.0 (18B) |
|---|---|---|---|
| Fineweb-Edu | 10B | 20B | 18B |
| DCLM | 0 | 0 | 2B |
| Cosmopedia V2 | 0 | 0 | 1B |
| OpenHermes 2.5 | 450M | 450M | 450M |
| **Total** | 10.5B | 20.5B | 21.5B |

*Table 10.* Open-source datasets used in our experiments.

| Dataset | Huggingface Data ID |
|---|---|
| Fineweb-Edu | `HuggingFaceTB/smollm-corpus/fineweb-edu-dedup` |
| Cosmopedia V2 | `HuggingFaceTB/smollm-corpus/cosmopedia-v2` |
| DCLM | `mlfoundations/dclm-baseline-1.0` |
| OpenHermes 2.5 | `teknium/OpenHermes-2.5` |
| Open-Orca/SlimOrca | `Open-Orca/SlimOrca` |
| Nemotron-Pretraining | `nvidia/Nemotron-Pretraining-Dataset-sample` |

### F.3. Model Configurations

The architecture of our RED student models follows the design of the LRC series. Table 11 presents a comprehensive comparison of the architectural configurations between the student models and their respective teachers. Specifically, each student model inherits the structural hyperparameters of its teacher (e.g., number of layers, attention heads, and FFN dimensions), with the primary modification being a reduction in the hidden size to approximately half that of the teacher.

*Table 11.* Architectural configurations of student models (RED series) and their respective teacher models.

| Configuration | 1.5B | | 1.7B | | 4B | |
|---|---|---|---|---|---|---|
| | RED-1.5B-Base | Llama3.2-3B-Ins. | RED-1.7B-Base | Qwen2.5-3B-Ins. | RED-4B-Base | Qwen2.5-7B-Ins. |
| Layers | 28 | 28 | 36 | 36 | 28 | 28 |
| Attn Q Heads | 24 | 24 | 16 | 16 | 28 | 28 |
| Attn KV Heads | 8 | 8 | 2 | 2 | 4 | 4 |
| Head Dim | 128 | 128 | 128 | 128 | 128 | 128 |
| Hidden Size | 1,536 | 3,072 | 1,200 | 2,048 | 2,048 | 3,584 |
| FFN Intermediate Size | 8,192 | 8,192 | 11,008 | 11,008 | 18,944 | 18,944 |
| Vocab Size | 128,256 | 128,256 | 151,936 | 151,936 | 152,064 | 152,064 |
| Tie Word Embeddings | True | True | True | True | False | False |

## F.4. Model Checkpoints

Table 12 summarizes the teacher and baseline model checkpoints utilized in our study. If the models in our experiments have weights available on Hugging Face, these will be prioritised for use. If not, the model will be trained according to the official code.

*Table 12.* Summary of model checkpoints and their Hugging Face ID.

| Model Name | Hugging Face Model ID |
|---|---|
| Llama3.2-3B-Instruct | `meta-llama/Llama-3.2-3B-Instruct` |
| Qwen2.5-3B-Instruct | `Qwen/Qwen2.5-3B-Instruct` |
| Qwen2.5-7B-Instruct | `Qwen/Qwen2.5-7B-Instruct` |
| Llama3.1-8B-Instruct | `meta-llama/Llama-3.1-8B-Instruct` |
| Llama3.1-8B-Base | `meta-llama/Llama-3.1-8B` |
| LLMStreamline-5.4B | `XiaodongChen/Llama-3.1-5.4B` |
| LLMStreamline-4.7B | `XiaodongChen/Llama-2-4.7B` |
| LRC-1.5B | `JitaiHao/LRC-1.5B-Base` |
| LRC-1.7B | `JitaiHao/LRC-1.7B-Base` |
| LRC-1.5B-SFT | `JitaiHao/LRC-1.5B-SFT` |
| LRC-1.7B-SFT | `JitaiHao/LRC-1.7B-SFT` |
| Gemma3-1B | `google/gemma-3-1b-pt` |
| Gemma3-1B-Instruct | `google/gemma-3-1b-it` |
| InternLM2-1.8B | `internlm/internlm2-1_8b` |
| InternLM2-1.8B-Instruct | `internlm/internlm2-chat-1_8b` |
| SmolLM2-1.7B | `HuggingFaceTB/SmolLM2-1.7B` |
| SmolLM2-1.7B-Instruct | `HuggingFaceTB/SmolLM2-1.7B-Instruct` |
| MiniCPM-1.2B | `openbmb/MiniCPM-1B-sft-bf16` |
| Llama-3.2-1B | `meta-llama/Llama-3.2-1B` |
| Llama-3.2-1B-Instruct | `meta-llama/Llama-3.2-1B-Instruct` |
| Minitron-4B | `nvidia/Minitron-4B-Base` |
| Minitron-4B-Instruct | `nvidia/Nemotron-Mini-4B-Instruct` |
| Gemma3-4B | `google/gemma-3-4b-pt` |
| Gemma3-4B-Instruct | `google/gemma-3-4b-it` |
| Llama-2-7b | `meta-llama/Llama-2-7b-hf` |
| Llama-2-7b-chat | `meta-llama/Llama-2-7b-chat-hf` |
| Llama-3.2-3B | `meta-llama/Llama-3.2-3B` |
| LRC-4B | `JitaiHao/LRC-4B-Base` |
| LRC-4B-SFT | `JitaiHao/LRC-4B-SFT` |

## F.5. Detailed Benchmark Descriptions and Evaluation Settings

All evaluations are conducted using the lm-evaluation-harness with transformers as the inference backend. To ensure reproducibility, we maintain the following default configurations: GSM8K is evaluated in a 5-shot setting, MBPP in a 3-shot setting, and all other benchmarks are conducted under 0-shot conditions. Below, we provide detailed descriptions of the benchmarks categorized by the core ability they assess and the reported metrics. It is important to distinguish the evaluation paradigms: tasks under general ability are formulated as discriminative multiple-choice questions, whereas multi-step reasoning tasks require the model to generate *generative* intermediate reasoning chains alongside the final answer.

**General Ability.** This category assesses the model's foundational language understanding, including factual knowledge acquisition, commonsense reasoning, and question answering over daily and scientific contexts. An overview of the benchmarks and evaluation metrics is summarized in Table 13.

**Scientific Understanding and Reading Comprehension.** These benchmarks probe the model's foundational ability to interpret scientific concepts and extract factual information from structured contexts.

*Table 13.* Evaluation Benchmarks and Metrics

| Category | Subcategory | Benchmark | Metric |
|---|---|---|---|
| **General Ability** | Scientific Understanding and Reading Comprehension | ARC-E/C (Clark et al., 2018) | Acc Norm |
| | | BoolQ (Clark et al., 2019) | Acc Norm |
| | Commonsense Understanding | PIQA (Bisk et al., 2020) | Acc Norm |
| | | WinoGrande (Sakaguchi et al., 2021) | Acc |
| | | HellaSwag (Zellers et al., 2019) | Acc Norm |
| | World Knowledge & Truthfulness | MMLU (Hendrycks et al., 2020) | Acc |
| | | TruthfulQA (Lin et al., 2022) | MC2 |
| **Multi-step Reasoning** | Mathematical Reasoning | GSM8K (Cobbe et al., 2021) | Exact Match |
| | Code Generation | HumanEval (Chen, 2021) | Pass@1 |
| | | MBPP (Austin et al., 2021) | Pass@1 |

- ARC-E/C (Clark et al., 2018): The AI2 Reasoning Challenge consists of elementary and challenge-level science questions. We report Acc Norm.

- BoolQ (Clark et al., 2019): A reading comprehension dataset for yes/no questions, evaluating fundamental language understanding. We report Acc Norm.

**Commonsense Understanding.** These benchmarks evaluate the model's grasp of everyday situations, physical interactions, and implicit human knowledge, which are essential for generating coherent and natural responses.

- PIQA (Bisk et al., 2020): Evaluates physical commonsense reasoning. We report Acc Norm.

- WinoGrande (Sakaguchi et al., 2021): A benchmark for commonsense reasoning with pronoun resolution. We report Acc.

- HellaSwag (Zellers et al., 2019): Challenges models to predict the most plausible continuation of a sentence. We report Acc Norm.

**World Knowledge and Truthfulness.** These datasets measure the breadth, depth, and reliability of the model's factual knowledge across diverse domains, with an emphasis on resisting common misconceptions and hallucinated content.

- MMLU (Hendrycks et al., 2020): A massive multitask language understanding benchmark covering 57 subjects. We report Acc.

- TruthfulQA (Lin et al., 2022): Designed to evaluate the truthfulness of model responses. We adopt the MC2 metric.

**Multi-step Reasoning.** This category evaluates the model's ability to perform multi-step logical reasoning and to synthesize structured and functional outputs. The evaluated tasks span mathematical problem solving and code generation, as detailed in Table 13.

**Mathematical Reasoning.** This benchmark assesses numerical reasoning over natural language descriptions, where correct solutions require multiple intermediate reasoning steps.

- GSM8K (Cobbe et al., 2021): A benchmark of high-quality grade-school math word problems. We use the Exact Match metric, with final numerical answers parsed using flexible-extract.

**Code Generation.** These benchmarks evaluate the model's ability to synthesize correct and executable programs from natural language specifications.

- HumanEval (Chen, 2021) and MBPP (Austin et al., 2021): Python programming benchmarks evaluated using Pass@1.

# G. Extra Ablations

## G.1. Training Dynamics and Convergence Analysis

To investigate the stability and efficiency of our distillation process, we analyze the performance evolution of LRC-1.5B-Base and RED-1.5B-Base across the entire training trajectory. Both models are trained on a total budget of 10 billion tokens. We evaluate checkpoints at four representative stages to monitor the acquisition of general knowledge (MMLU) and the preservation of multi-step reasoning (GSM8K).

The detailed evaluation results are presented in Table 14. While both models exhibit a similar and steady upward trend on MMLU, eventually converging to a score of 0.50, the reasoning trajectory of LRC-1.5B-Base is significantly lagged and suppressed. In contrast, RED-1.5B-Base maintains a robust and monotonic improvement in both metrics, suggesting that our method avoids the reasoning collapse. Notably, at only 2.88B tokens, RED-1.5B-Base achieves a GSM8K score of 0.27, which already outperforms the final converged performance of LRC-1.5B-Base (0.04 at 10B tokens). It is worth noting that the LRC-1.5B-Base and RED models reported here are trained on the same codebase environment and data. KL temperature of LRC is set as 40, which is consistent with its original setting. However, their GSM8K results are lower than those of the official checkpoint ($0.21 \rightarrow 0.04$). The results presented in the main text correspond to the official checkpoint.

*Table 14.* Detailed performance of LRC-1.5B-Base vs. RED-1.5B-Base over 10B training tokens. Metrics reported are MMLU and GSM8K. Our model (**RED**) consistently exhibits higher reasoning throughout the training duration.

| Training Tokens | Steps | LRC-1.5B-Base | | RED-1.5B-Base (Ours) | |
|---|---|---|---|---|---|
| | | MMLU | GSM8K | MMLU | GSM8K |
| 2.88 Billion | 60k | 0.44 | 0.05 | 0.44 | **0.27** |
| 6.73 Billion | 140k | 0.49 | 0.05 | 0.49 | **0.39** |
| 8.65 Billion | 180k | 0.49 | 0.13 | 0.49 | **0.42** |
| 10.00 Billion | 208k | 0.50 | 0.04 | **0.50** | **0.44** |

## G.2. Robustness Analysis of Calibration Datasets

In this section, we investigate the sensitivity of our channel-selection mechanism to the choice of calibration datasets and their respective sample sizes. Our empirical findings demonstrate two key properties: (i) the subset of identified "important" channels exhibits a high degree of overlap across different data distributions and sample scales, and (ii) the selection process is remarkably sample-efficient.

Critically, unlike traditional pruning-and-retraining methods that statically discard weights from the teacher model, our approach utilizes the selected channel indices solely to initialize the projection matrices. Since these matrices remain learnable and are updated during the subsequent distillation process, the performance of the student model is less coupled to the initial calibration set.

*Table 15.* Overlap analysis of selected channels for RED-1.5B under various calibration settings. The baseline is highlighted (Nemotron, $N = 15$). The high overlap ratios across different datasets and sizes demonstrate the robustness of our selection method.

| Calibration Dataset | Sample Size ($N$) | Overlap Count | Overlap Ratio (%) |
|---|---|---|---|
| Nemotron-Pretraining | 3 | 1,295 | 84.31% |
| | 9 | 1,429 | 93.03% |
| | **15 (Baseline)** | **1,536** | **100.00%** |
| | 30 | 1,452 | 94.53% |
| SlimOrca | 3 | 1,212 | 78.91% |
| | 9 | 1,251 | 81.45% |
| | 15 | 1,268 | 82.55% |
| | 30 | 1,298 | 84.51% |

Table 15 presents the overlap of selected channels for the RED-1.5B model, where we select the top $1,536$ dimensions out of a $3,072$ hidden dimension space. Taking our default setting (15 samples from the Nemotron-Pretraining dataset) as the

baseline, we compare the Jaccard similarity when varying the dataset to SlimOrca and scaling the number of samples from 1 to 30. The results indicate that even with as few as 3 samples, the overlap ratio remains above 80%, and the core features identified are remarkably consistent across different data distributions, justifying the robustness of our initialization strategy.

### G.3. Using Base Model As the teacher

We compare the performance of LRC, RED and LLMStreamline when all utilise Llama3.1-8B-Base as the teacher model, as shown in Table 16. Compared to LRC, RED still exhibits a significant recovery in multi-step reasoning capability, while demonstrating a strong performance advantage over LLMStreamline, which is a depth-reduced Distill method.

*Table 16.* Performance comparison with the same Base model as the teacher

| Paradigm | EDistill | | |
|---|---|---|---|
| **Model** | **LLMStr.** | **LRC** | **RED (Ours)** |
| **(Size)** | **(4.0B)** | **(4.0B)** | **(4.0B)** |
| Teacher | Llama3.1-8B-Base | Llama3.1-8B-Base | Llama3.1-8B-Base |
| # Tokens | 1.3B | 18B | 18B |
| Dataset | SlimPajama | Mixed-2.0 | Mixed-2.0 |
| *General Ability* | | | |
| MMLU | 0.23 | 0.50 | **0.52** |
| PIQA | 0.71 | 0.75 | **0.78** |
| ARC-E | 0.52 | 0.72 | **0.75** |
| ARC-C | 0.28 | 0.44 | **0.47** |
| BoolQ | 0.48 | 0.74 | **0.75** |
| WinoGrande | 0.51 | 0.65 | **0.68** |
| HellaSwag | 0.46 | 0.68 | **0.72** |
| TruthfulQA | 0.40 | 0.43 | 0.43 |
| **Gen. Avg.** | 0.45 | 0.61 | **0.64** |
| *Multi-step Reasoning (Math & Code)* | | | |
| GSM8K | 0.01 | 0.15 | **0.27** |
| MBPP | 0.00 | 0.13 | **0.21** |
| HumanEval | 0.01 | 0.05 | **0.16** |
| **Reas. Avg.** | 0.01 | 0.11 | **0.21** |

### G.4. Benchmarking Base vs. Instruct Models

In our primary experiments, we prioritize the comparison of Base models over their Instruct counterparts. This is because the performance of modern Instruct models is increasingly driven by specialized data mixtures. A common industry practice is to incorporate substantial amounts of domain-specific data, particularly **mathematical reasoning and programming code**, during the SFT or RLHF stages to improve benchmark scores. Moreover, reinforcement learning (RL) is often employed when training Instruct models, which further enhances multi-step reasoning ability. A prominent example is Gemma3-4B, which sees its GSM8K performance nearly double ($0.38 \rightarrow 0.76$) after instruction tuning, likely due to such targeted data enrichment.

By focusing on *Base* models, we ensure a more controlled assessment of the architectural and distillation-induced strengths. Although our RED-Base models are distilled from a high-capacity Instruct teacher, they are evaluated here as foundational initializations without undergoing any further supervised fine-tuning (SFT) or RL. As shown in Table 17, our **RED-Base** models (1.5B and 4B) already outperform several Instruct models in both general knowledge (MMLU) and reasoning (GSM8K).

*Table 17.* Comparison of Base and Instruct versions. Scores are reported for MMLU and GSM8K. The variance in Δ GSM highlights how instruction tuning recipes (often involving heavy math/code data) can lead to inconsistent trends across model series.

| Model Family | Base (MMLU/GSM8K) | Instruct (MMLU/GSM8K) | Δ MMLU | Δ GSM8K |
| --- | --- | --- | --- | --- |
| InternLM2-1.8B | 0.41 / 0.23 | 0.43 / 0.34 | +0.02 | +0.11 |
| Gemma3-1B | 0.24 / 0.03 | 0.39 / 0.25 | +0.15 | +0.22 |
| Gemma3-4B | 0.58 / 0.38 | 0.58 / 0.76 | 0.00 | +0.38 |
| Minitron-4B | 0.56 / 0.25 | 0.57 / 0.31 | +0.01 | +0.06 |
| **RED-1.5B (Ours)** | **0.51 / 0.44** | - | - | - |
| **RED-4B (Ours)** | **0.62 / 0.49** | - | - | - |

## G.5. Irreversibility of eRank and Reasoning Collapse

We investigate whether standard instruction tuning can mitigate the *eRank collapse* and *reasoning collapse* observed in specific distilled models, such as the LRC series.

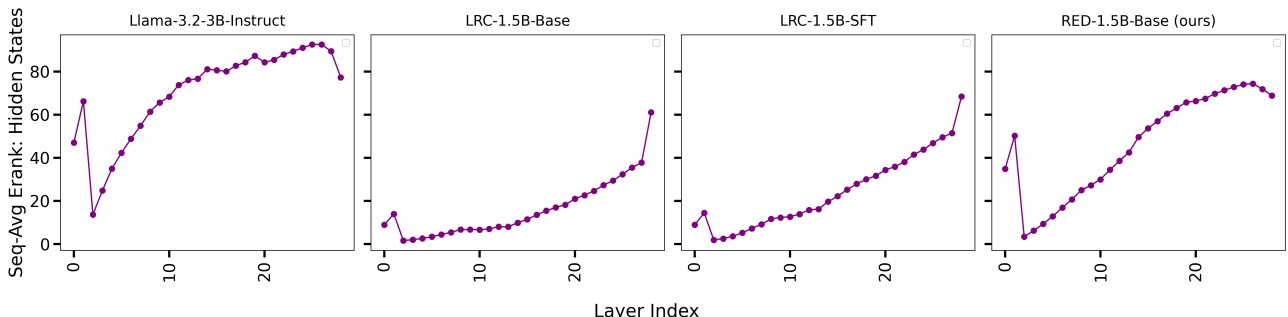

*Figure 6.* Comparison of eRank across different layers.

As a case study, we evaluate the LRC-1.5B-Base model before and after SFT on 0.2B tokens of the Ultra-chat dataset. As illustrated in Figure 6, the SFT process fails to recover the model's representation quality; the eRank of the SFT version remains significantly lower than that of the teacher model. Quantitatively, we observe a degradation in mathematical reasoning: the GSM8K accuracy drops from 0.21 (Base) to 0.18 (SFT). This suggests that general-purpose conversational SFT is insufficient to resolve underlying *eRank collapse* and *reasoning collapse*. In contrast, our **RED-1.5B-Base** model, through optimized initialization, inherently maintains a high eRank and achieves a GSM8K score of 0.44 without specialized RL or mathematical fine-tuning. This evidence supports the hypothesis that the reasoning collapse of LRC is deeply rooted in the base distillation phase and may be irrecoverable if lost at that stage.

## G.6. Effects of Different Importance Estimation Strategies.

We further investigate the sensitivity of our RED framework to various importance estimation strategies. Our default strategy is based on the mean absolute magnitude of hidden representations. We compare this with two alternative strategies: (i) the Minitron-style approach, which calculates magnitudes at the output of each normalization layer, and (ii) a QR-based (Antil et al., 2018) strategy that selects principal components from hidden representations (see Appendix H for technical details). As summarized in Figure 7, the performance of RED-1.5B with all three strategies is remarkably consistent, with MMLU scores ranging from 0.50 to 0.51 and GSM8K scores from 0.44 to 0.45. This marginal variance demonstrates the robustness of our RED method, indicating that its efficacy in width reduction is not strictly tied to a specific importance metric.

# H. Details of Other Importance Estimation Strategies

To further validate our approach, we consider two alternative importance estimation strategies. Both methods utilize the same calibration dataset $\mathcal{D}_{\text{pre}}$.

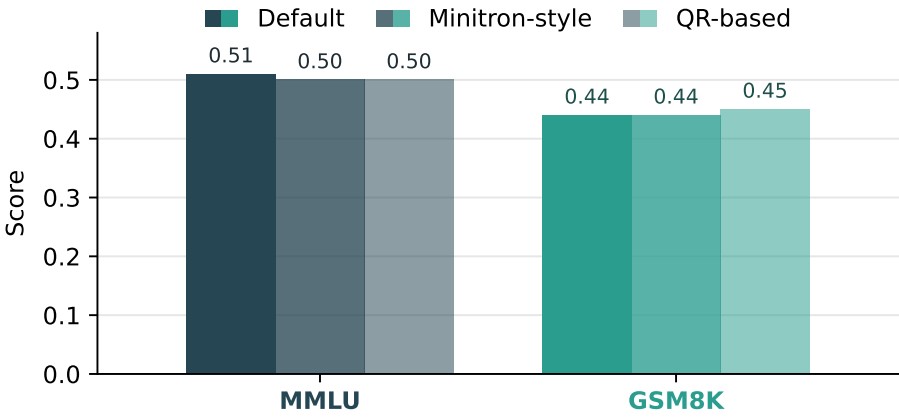

*Figure 7.* Comparison of Importance Estimation Strategies.

## H.1. QR-Decomposition-Based Importance Estimation

This strategy treats the importance estimation as a column subset selection problem, aiming to identify channels that are most linearly independent and capture the maximum variance of the activation space.

For each layer $i$, we first concatenate the hidden representations $X_i^{(k)}$ for all sequences $k = 1, \ldots, |\mathcal{D}_{\text{pre}}|$ along the sequence dimension to form a large activation matrix $\mathbf{A}_i \in \mathbb{R}^{(|\mathcal{D}_{\text{pre}}| \cdot L) \times D}$, where each column represents the activations of a specific channel across all tokens in the calibration set.

We then perform QR decomposition with column pivoting on $\mathbf{A}_i$. This algorithm iteratively selects the column that has the largest residual norm relative to the subspace spanned by the previously selected columns. The physical meaning of this process is captured by the resulting upper triangular matrix $\mathbf{R}_i$: the absolute value of its $j$-th diagonal element, $|\mathbf{R}_i^{jj}|$, represents the "unique contribution" of the $j$-th selected channel. A larger value indicates that the channel contains significant information that cannot be linearly represented by other already-selected channels.

After the decomposition, we assign these diagonal absolute values as the importance scores for their corresponding original channel indices, yielding the layer-wise scores $S_{i,j}$. These scores are then aggregated across all $N$ layers to compute the global importance $\bar{s}_j$ and determine the final index set $G$, following the same aggregation and top-indices selection logic described in the main text.

## H.2. Mintron-Style Activation Estimation

The second alternative follows the Mintron-style approach. The mathematical formulation of this strategy is identical to the *Activation-Based Importance Estimation* described in the main text, with the sole distinction being the location of the activation. Specifically, while our default method monitors the input and output of each Transformer layer, the Mintron-style method computes the importance scores based on the outputs of each normalization layer (e.g., RMSNorm) within the Transformer layer. This strategy aims to capture the importance of features after they have been re-centered and re-scaled by the normalization statistics.

# I. Extra Definition

## I.1. RMSNorm

The RMSNorm function for a generic input vector $x \in \mathbb{R}^D$ at a specific layer component $k$ is defined as:

$$\text{RMSNorm}_k(x) = \frac{x}{\sqrt{\frac{1}{D}\|x\|_2^2 + \epsilon}} \odot g_k. \tag{87}$$

where $g_k \in \mathbb{R}^D$ is the learnable scaling parameter unique to that normalization layer, and $\epsilon$ is a small constant that ensures numerical stability.

### I.2. Calculation of Token Entropy

To measure the predictive uncertainty across different tokens, we define the **Token Entropy** based on the output probability distributions. Recall that $X \in \mathbb{R}^{L \times D}$ is the representation matrix for a sequence of length $L$. For the $l$-th token ($l \in \{1, \ldots, L\}$), its feature vector is denoted as $X^l \in \mathbb{R}^D$. The corresponding probability distribution $P^l \in \mathbb{R}^{\text{Voc}}$ is generated via the unembedding layer:

$$P^l = \text{softmax}\left(\text{Norm}_{\text{final}}(X^l)W_u\right), \tag{88}$$

where $W_u \in \mathbb{R}^{D \times \text{Voc}}$ is the unembedding weight matrix.

Specifically, the Shannon entropy for the $l$-th token, denoted as $H(P^l)$, is calculated as:

$$H(P^l) = -\sum_{j=1}^{\text{Voc}} P_j^l \log P_j^l, \tag{89}$$

where $P_j^l$ represents the probability of the $j$-th position in the vocabulary. For a single sequence, we compute the average entropy across all $L$ tokens:

$$\bar{H} = \frac{1}{L} \sum_{l=1}^{L} H(P^l). \tag{90}$$

In our experiments, we sample a set of sequences. The final reported Token Entropy is defined as the mean value across all sampled sequences. This metric reflects the average uncertainty of the model's predictions.

