# OpenReview forum: "Reasoning-preserved Efficient Distillation of Large Language Models via Activation-aware Initialization"
_ICML.cc/2026/Conference — ICML 2026 regular_

### Official Review · Reviewer_NVta · 2026-03-08

**Soundness:** 3
**Presentation:** 2
**Significance:** 3
**Originality:** 3
**Overall Recommendation:** 4
**Confidence:** 3

**Summary:**

This paper proposes RED, an efficient distillation method for large language models (LLMs) that replaces random initialization with activation-aware initialization for width-reduced projection matrices. The authors provide a theoretical analysis demonstrating that randomly initialized projections can lead to uneven singular value growth, resulting in a substantial reduction in the effective rank of hidden representations and increased token indistinguishability, ultimately impairing reasoning capabilities. Empirically, RED significantly improves multi-step reasoning performance (e.g., on GSM8K, MBPP, and HumanEval) while preserving general capabilities and maintaining high training efficiency across models ranging from 1.5B to 4B parameters.

**Compliance With Llm Reviewing Policy:**

Affirmed.

**Final Justification:**

The rebuttal have addressed my concerns. I think the strengths outweigh the weaknesses. Thus, I choose to maintain my score.

**Key Questions For Authors:**

1. Why reduced eRank does not affect general capabilities? Could authors provide more analysis on it?
2. In the implementation, the same selection matrix H is utilized for all projection matrices to preserve activation subspace. What if different H is used for each projection matrix? Will it greatly affect reasoning performance after distillation?
3. As shown in LRC paper, we use different projection matrices for gate and up layer. But in Equation 3, it seems that RED uses single project matrix $O^f$ for gate and up layer. Are there any effects for the final results? Moreover, it's suggested that the authors can add some formulation about the final weight matrices (i.e., weight matrices of teacher model with projection) for student model for better clarity.

**Limitations:**

See in weaknesses and questions.

**Strengths And Weaknesses:**

Strengths:
1. The paper is well structured.
2. Extensive theoretical analysis is provided to demonstrate the shortcomings of random initialization for projection matrices in width-reduced efficient distillation.
3. A new activation-aware initialization is introduced to address the eRank collapse issue of hidden representations.


Weaknesses:
1. The theoretical analysis of gradient is based on a simple linear auto encoder proxy, which may have a gap with real LLM dynamics which is much complicated.
2. The methodology primarily focuses on width-reduced based efficient distillation, which is a relatively narrow area. If this kind of initialization method can be extended to other scenarios, the influence will be greater.

---

> ### Author Rebuttal · Authors · 2026-03-30
>
> We sincerely thank the reviewer for the thoughtful and constructive comments. We address each concern below.
>
> > W1: Linear proxy
>
> We agree that the linear autoencoder proxy does not fully capture nonlinear LLM dynamics. However, Theorem 4.2 makes a concrete **early-training prediction**: larger singular values should grow faster. We directly test this in real LRC training by correlating singular values $\sigma$ with their empirical growth $\Delta\sigma$.
> | Interval | Up Proj. $r$ | Down Proj. $r$ | Phase |
> | :--- | :---: | :---: | :--- |
> | 20k $\rightarrow$ 40k | 0.8948 | 0.5454 | Early |
> | 60k $\rightarrow$ 80k | 0.0057 | -0.6936 | Transition |
> | 180k $\rightarrow$ 200k | -0.6703 | -0.9388 | Late |
>
> **These results provide strong quantitative support in the early phase: larger singular values indeed grow faster, driving rapid spectral imbalance from small random initialization**. As training progresses, nonlinearities and architectural bounds limit further divergence, but by then eRank collapse has already occurred. We also track the normalized singular value sum of Down matrix, $Nesum = \sum_i (\sigma_i/\sigma_{\max})$, which measures spectral uniformity.
>
> | Step | RED ($Nesum$) | LRC ($Nesum$) |
> | :--- | :---: | :---: |
> | Init | 1536 | 840 |
> | 20k | 808 | 397 |
> | 180k | 549 | 310 |
> | 200k | 551 | 309 |
>
> **Despite nonlinearities, RED still stabilizes at a much richer spectrum than LRC.**
>
> > W2:Narrow scope
>
> We agree that the current paper focuses on width-reduced EDistill. However, the core principle behind RED is more **general**: activation-aware initialization to preserve important subspaces when new projection layers are introduced. For instance, in multimodal alignment (e.g., VLMs), frozen LLMs are combined with new projection layers for modality alignment. RED may be employed in such settings, and we leave this as future work.
>
> > Q1:Reduced eRank does not affect general capabilities
>
> Thank you for this question. For **general tasks (e.g., MMLU)**, the model mainly needs to preserve the correct Top-1 ranking among a small set of answer candidates, so blurrier representations can still yield the correct choice. In contrast, **multi-step reasoning (e.g., GSM8K)** requires preserving and updating intermediate states over a long trajectory. When eRank collapses, these states become less separable.
>
> We analyze the following GSM8K example:
>
> ***Prompt:* "A tree grows 4 inches every month. If the tree is currently 5 feet tall, how tall will it be in 2 years? (1 foot = 12 inches)"**
> - **RED (Ours, Succeeded):** `5 * 12 = 60`; `4 * 12 = 48`; `48 * 2 = 96`; finally, `the tree will be 60 + 96 = 156 inches tall.` RED preserves both key states, `60` and `96`, until the final composition step.
> - **LRC (Failed):** `5 * 12 = 60`; `48 + 48 = 96`; but then `the tree will be 96 * 12 = 1,192 inches tall.` The key error is the final composition step: LRC loses the original height `60` and wrongly treats `96` as a quantity to convert again, instead of adding `60` and `96`.
>
> This is also reflected in the internal dynamics. The average representation similarity across reasoning steps is **0.40** for LRC but only **0.13** for RED. For the key phrase `the tree will be`, the **nearest tokens** of `be` in LRC are `48`, `12`, and `4`, while the crucial state `60` is missing; in RED, the nearest tokens already include `60`. **At the final operator decision**, LRC has entropy **1.28**, with top probabilities `* (0.5)`, `/ (0.2)`, and `+ (0.2)`, whereas RED has entropy **0.18**, with probabilities `+ (0.97)` and `inches (0.03)`, showing a much sharper distribution.
>
> **This case study is consistent with Theorems 4.1-4.3: LRC exhibits much higher inter-step similarity, and remains uncertain at the final operator step**
>
> > Q2:Different selection matrices $H$
>
> This is an insightful suggestion. Our current design deliberately uses a **shared selection matrix $H$** across projection matrices to preserve **channel consistency** throughout the residual stream. Since hidden channels are repeatedly reused and combined through residual connections, **using different $H$ matrices for different modules may introduce misalignment of channel semantics across layers and harm preservation of the pretrained feature space**. Due to rebuttal-time limits, we have not yet conducted layer-specific  $H$ experiments, but we will add this in the revised revision.
>
> > Q3:Eq. 3 seems to use a single projection for Gate/Up
>
> We thank the reviewer for catching this presentation issue. The notation in Eq. 3 is overly simplified. Our actual implementation is identical to LRC in this respect: we use independent learnable projection matrices for the Gate and Up layers, as also stated in Appendix C. **Therefore, this is a notation issue only and does not affect the reported results**. In the revision, we will rewrite Section 3 to distinguish these projections and show the final merged weight formulation more clearly.

---

> > ### Author Rebuttal · Reviewer_NVta · 2026-04-02
> >
> > Thanks for your detailed response. I would like to maintain my score.

---

> > > ### Author Response · Authors · 2026-04-03
> > >
> > > Thank you for your positive feedback and valuable suggestions. We appreciate the time and effort you have put in.

---

### Official Review · Reviewer_QoWh · 2026-03-08

**Soundness:** 2
**Presentation:** 2
**Significance:** 2
**Originality:** 1
**Overall Recommendation:** 4
**Confidence:** 4

**Summary:**

This paper improves upon efficient distillation (EDistill) for compressing LLMs using projection-based width reduction. The authors observe what they call reasoning collapse, where distilled models maintain general benchmark performance but perform poorly on multi-step reasoning tasks. They attribute this to effective rank (eRank) collapse caused by using randomly initialized projection matrices. The proposed method (RED) replaces random initialization with activation-aware channel selection based on a small calibration dataset. The study's major finding is that this initialization improves reasoning benchmarks while maintaining efficiency.

**Compliance With Llm Reviewing Policy:**

Affirmed.

**Final Justification:**

My main concerns were over the accuracy differences and some missing experiments to highlight why RED is more appropriate over types of orthogonalization to preserve information and avoid collapse. I read through the other reviewer comments carefully and I have no more additional questions.

**Key Questions For Authors:**

Why do some baseline numbers differ from those reported in the original LRC paper?
How does RED compare to orthogonal projection methods or orthogonality regularization?
Why does initialization appear to impact reasoning benchmarks more strongly than general benchmarks? This is very important, since reasoning is the main narrative of this paper.

**Limitations:**

The limitations discussed in the supplementary are very broad (like not having RL experiments). I think these are not the limitations of this work and would be more future work. Although I do not think there are any direct negative societal impacts of this work, this was not mentioned in the paper explicitly.

**Strengths And Weaknesses:**

Weaknesses
The core contribution is mainly an initialization strategy rather than a fundamentally new distillation method.
The term EDistill is relatively new and should be introduced more clearly before appearing in the abstract (for example a preliminary section before the method).
It would be useful to discuss the relation to orthogonal projection approaches such as VkD [1]. A comparison to orthogonal projections or orthogonality regularisation [2] would be an important ablation. Using the same theory as [3] to structure the analysis is nice and it logically flows into the two EDistill settings. However, it does raise the question as to why the ideas from [3] cannot also be applied to EDistill? to strengthen the originality of the findings in this paper, it should be clear why this initialisation is specific to EDistill as well.

Some numbers reported in Table 3 appear inconsistent with those reported in [4]. For example ARC-C (LRC 1.5B) (45% v.s. 42%) and ARC-E (75% v.s. 70%) appear lower here than in the original paper. Since the reported average improvements are small (~2%), clarification would be helpful. The original EDistill paper does not report reasoning results, s
Ablation experiments are somewhat limited. In particular, comparisons to orthogonal initialization, orthogonality regularization, and different channel-selection strategies would strengthen the paper.
The theoretical analysis using a linear autoencoder may be better suited to the supplementary, and further experimental analysis explaining why initialization particularly affects reasoning tasks is missing

[1] Improving Knowledge Distillation using Orthogonal Projections. CVPR 2024

[2] Preventing Dimensional Collapse in Self-Supervised Learning via Orthogonality Regularization. NeurIPS 2024

[3] Understanding Dimensional Collapse in Contrastive Self-supervised Learning. ICLR 2022

[4] A Token is Worth over 1,000 Tokens: Efficient Knowledge Distillation through Low-Rank Clone. NeurIPS 2025

Strengths
The paper finds an interesting settings where efficient distillation degrades on reasoning benchmarks despite strong general performance.
The proposed method is easy to implement and with little additional training costs.
Empirical improvements over the LRC baseline on reasoning tasks appear promising.
The theory seems to be logically correct and supplements the submission.

---

> ### Author Rebuttal · Authors · 2026-03-30
>
> We sincerely thank the reviewer for the thoughtful and constructive comments. We address each concern below.
>
> > W1:Mainly an initialization strategy rather than a new distillation method?
>
> We agree that RED is not a new distillation architecture. **Its contribution is a diagnosis-and-fix for a previously overlooked failure mode in EDistill: reasoning collapse**. We show that EDistill can preserve general benchmarks while degrading on multi-step reasoning, analyze this through eRank collapse of projection matrices, and derive a targeted activation-aware initialization that mitigates this issue while preserving efficiency.
>
> > W2:EDistill should be introduced more clearly.
>
> We will define EDistill more clearly in the abstract and add a short preliminary section. Concretely, EDistill refers to efficient distillation methods that reduce depth or width while freezing inherited teacher LLM parameters and optimizing only lightweight inserted modules.
>
> > **W3,Q2:Comparison to orthogonal projection / orthogonality regularization is needed.**
>
> We totally agree to conduct this comparison. We only use VkD’s orthogonal initialisation [1], as bf16 does not support orthogonal operators during training. The Frobenius norm is adopted as the orthogonal regulariser [2]. The comparisons of them in both the autoencoder and Edistill settings are carried out.
>
> | Method | Recon. $\downarrow$ | eRank $\uparrow$ |
> | --- | ---: | ---: |
> | Random | 1.24 | 67.06 |
> | Random + OR (0.001) | 1.24 | 111.82 |
> | Random + OR (0.01) | 1.52 | 172.60 |
> | Random Orth. Init | 0.50 | 149.97 |
> | Ours | 0.08 | 263.28 |
>
> | Method | MMLU | GSM8K |
> | --- | ---: | ---: |
> | LRC (Random) | 0.50 | 0.21 |
> | LRC + Random Orth. Init | 0.50 | 0.30 |
> | LRC + OR (1e-4) | 0.48 | 0.26 |
> | RED | 0.51 | 0.44 |
>
> These results show that generic orthogonal initialization/regularization improves eRank, but still starts from random singular directions and underperforms RED in both reconstruction and reasoning. **This also clarifies where RED differs from generic orthogonal methods: RED not only encourages a healthier spectrum, but also preserves important teacher subspaces at initialization**. We will add this important comparison in the revision. Thanks again for your valuable suggestions.
>
> > W4:Why can [3] not be directly applied to EDistill?
>
> **[3] studies collapse under InfoNCE, while our analysis is for the MSE-style reconstruction objective used in EDistill**. Its remedy, removing the projector and imposing whitening/contrastive constraints, is therefore not directly compatible with standard EDistill pipelines, which rely on frozen-teacher reconstruction with inserted width/depth reduction modules. We will clarify this distinction more explicitly.
>
> > **W5,Q1:Some Table 3 numbers appear inconsistent with LRC.**
>
> Thanks for the point! **We assure that we did not lower the baseline**. The apparent discrepancy comes from two factors: (1) **checkpoint mismatch**: the 45%/75% results in the LRC main text are for the SFT model, while RED is a Base model and should be compared with LRC-Base; (2) **metric mismatch**: LRC reports ARC-E with Acc and ARC-C with Acc\_norm, whereas we use Acc\_norm for both for consistency across benchmarks. We will clarify these in the revised version.
>
> | Model | ARC-E (A) | ARC-E (A_N) | ARC-C (A) | ARC-C (A_N) | GSM8K |
> | --- | ---: | ---: | ---: | ---: | ---: |
> | LRC-Base (Paper) | 0.73 | - | - | 0.42 | - |
> | LRC-SFT (Paper) | 0.75 | - | - | 0.45 | - |
> | LRC-Base (Ours) | 0.74 | 0.70 | 0.39 | 0.42 | 0.21 |
> | LRC-SFT (Ours) | 0.75 | 0.72 | 0.42 | 0.45 | 0.18 |
>
>
> > W6:Different channel-selection strategies would strengthen the paper.
>
> We already conducted this ablation in Appendix G.6 and will move it to a more prominent place.
>
> | Strategy | MMLU | GSM8K |
> | --- | ---: | ---: |
> | RED (Default) | 0.51 | 0.44 |
> | RED (Minitron-style) | 0.50 | 0.44 |
> | RED (QR-based) | 0.50 | 0.45 |
>
> The results are highly consistent across strategies, suggesting that **the benefit comes from orthogonal initialization plus preserving critical channels**, rather than from one specific selection rule.
>
> > W7,Q3:linear autoencoder may be better suited,  why initialization particularly affects reasoning.
>
> We agree. We will move the linear-autoencoder setup details to the appendix and use the saved space for a more direct empirical explanation of the reasoning effect.
>
> Our current hypothesis is that GSM8K-like tasks are more sensitive because they require autoregressive multi-step generation and retaining more independent intermediate states over long trajectories, whereas MMLU mainly requires preserving the correct relative ranking among a small set of answer tokens. A GSM8K case study (**see Reviewer NVta**) supports this: LRC shows **higher inter-step representation similarity (0.40 vs. 0.13) and higher entropy on the key operation token (1.28 vs. 0.18) than RED**, which is consistent with Theorems 4.1-4.3 (erank collapse and token-indistinguishability).

---

> > ### Author Rebuttal · Reviewer_QoWh · 2026-04-01
> >
> > The authors have addressed my concerns. With the additional experiments against orthogonal projections and orthogonal initialisation, it is clear RED is an improved approach here. This would be good to see in the main paper. The authors have addressed my concern over the accuracy differences and I will update my rating accordingly.

---

> > > ### Author Response · Authors · 2026-04-02
> > >
> > > Thank you for your positive feedback and for the time and effort you’ve invested. We will definitely revise our paper based on your suggestions, which have greatly improved the quality of our work.

---

### Official Review · Reviewer_8tgS · 2026-03-12

**Soundness:** 3
**Presentation:** 3
**Significance:** 3
**Originality:** 3
**Overall Recommendation:** 4
**Confidence:** 4

**Summary:**

This paper investigates the severe degradation of multi-step reasoning capabilities—termed "reasoning collapse"—in Large Language Models compressed via Efficient Distillation (EDistill). The authors theoretically trace this failure in width-reduced models to "eRank collapse," proving that standard random initialization of projection matrices causes a few singular values to dominate the representation space, rendering tokens indistinguishable. To resolve this, they introduce Reasoning-preserved Efficient Distillation (RED), which utilizes an activation-aware initialization strategy based on a small calibration set. This initialization theoretically and empirically stabilizes the singular value dynamics, preventing dimensional collapse. Extensive experiments across Llama and Qwen architectures (1.5B to 4B parameters) demonstrate that RED achieves state-of-the-art general abilities and significantly recovers multi-step reasoning while maintaining extreme training efficiency compared to full-parameter recovery methods.

**Compliance With Llm Reviewing Policy:**

Affirmed.

**Final Justification:**

Added my final consideration in the comment, will keep my judgement

**Key Questions For Authors:**

1. Can you provide results where LRC is retrained with τ=1 using a properly tuned learning rate and other hyperparameters? If LRC fundamentally cannot train at τ=1, can you explain why theoretically — is this a consequence of random initialization specifically, or could there be other factors? How does RED perform at intermediate temperatures (e.g., τ=5, τ=10)?
2. Can you provide quantitative measurements of how closely the actual singular value evolution of projection matrices during RED/LRC training follows the predicted gradient flow dynamics (Theorem 4.2)? Specifically, is the exponential divergence rate for LRC quantitatively predicted by the theory?
3. Have you tried SFT or RL specifically targeting math/code reasoning on LRC models? The irreversibility experiment (G.5) only uses general conversational SFT. It's plausible that targeted reasoning training could close the gap, which would weaken the significance of the initialization-stage fix.
4. How does RED perform on more challenging benchmarks like MATH, BBH, or ARC-Challenge with chain-of-thought prompting? GSM8K is relatively easy and may not fully differentiate reasoning capabilities.

**Limitations:**

The authors discuss limitations in Appendix A, acknowledging the lack of specialized math/code training data and the absence of post-training evaluation (SFT/RL).

**Strengths And Weaknesses:**

**Strength**
1. Complete causal chain from initialization to reasoning failure - The paper constructs a full mechanistic pipeline: random init → uneven singular value growth → skewed projection matrix spectrum → eRank collapse of XQ → bounded token indistinguishability via TV distance (Theorem 4.3, Eq. 8) → reasoning degradation.
2. Good reasoning improvements
3. Convergence speed - Table 12 shows RED-1.5B reaches GSM8K 0.27 at only 2.88B tokens (60k steps), already exceeding LRC's final performance at 10B tokens. By 6.73B tokens RED is at 0.39.

**Weakness**
1. Gap between theory and practice - The theoretical analysis is conducted on a linear autoencoder proxy, which strips away nonlinearities, attention mechanisms, residual connections, and normalization — all of which interact with the projection matrices in the actual EDistill setup. While this is acknowledged, the paper does not quantify how much the theoretical predictions deviate from actual training dynamics. For instance, does the singular value evolution of actual projection matrices in RED/LRC during real training quantitatively match the predicted dynamics? Figure 4 shows qualitative alignment but not quantitative.
2.  Depth-reduced analysis is shallow - The paper identifies reasoning collapse in both depth-reduced and width-reduced EDistill but only provides a theoretical explanation and solution for the width-reduced case.

---

> ### Author Rebuttal · Authors · 2026-03-30
>
> We sincerely thank the reviewer for the thoughtful and constructive comments. We address each concern below.
>
> >Q1: LRC retrained at $\tau=1$? RED at intermediate temperatures?
>
> Thank you for raising this point. We retrain LRC-1.5B ($\tau=1$, lr=$5e-4, 5e-5$), but the KL penalty drop remains slow and fluctuating.
>
> | Model | Temp. | MMLU | GSM8K |
> | --- | --- | ---: | ---: |
> | LRC-1.5B | $\tau=1$ | 0.25 | 0.05 |
> | LRC-1.5B | $\tau=40$ | 0.50 | 0.04 |
> | RED-1.5B | $\tau=1$ | 0.50 | 0.44 |
> | RED-1.5B | $\tau=5$ | 0.50 | 0.32 |
> | RED-1.5B | $\tau=40$ | 0.52 | 0.08 |
>
> At $\tau=1$, the teacher distribution is highly sharp. **Under random initialization, LRC perturbs the residual stream substantially at the beginning of training, causing severe student-teacher logit misalignment. This produces a massive initial KL penalty and unstable early updates**. Empirically, LRC at $\tau=1$ starts at about 25,000 KL and only drops to about 450. In contrast, RED preserves the initial logits much better through activation-aware initialization, starting from about 12,121 and converging smoothly to about 300.
>
> **We also train RED at $\tau=5$**; compared with $\tau=1$, the softer targets reduce the constraint on reasoning-critical dark knowledge, and reasoning drops accordingly.
>
> > Q2,W1:Quantitative support?
>
> We appreciate the question. Theorem 4.2 predicts a winner-take-all dynamic in the early stage of training, namely $\dot{\sigma}_M^r \propto \sigma_M^r$. While the exact divergence rate is difficult to predict in the full nonlinear network, we can directly test its core prediction: whether the empirical growth rate ($\Delta\sigma$) is positively correlated with the singular value ($\sigma$) itself in early training (Calculating Pearson Cor).
>
> | Interval | Up Proj. | Down Proj. | Phase |
> | --- | ---: | ---: | --- |
> | 20k $\rightarrow$ 40k | 0.8948 | 0.5454 | Early |
> | 60k $\rightarrow$ 80k | 0.0057 | -0.6936 | Transition |
> | 180k $\rightarrow$ 200k | -0.6703 | -0.9388 | Late |
>
> **These results provide strong quantitative support in the early phase: larger singular values indeed grow faster, driving rapid spectral imbalance from small random initialization**. As training progresses, nonlinearities and architectural bounds limit further divergence, but by then, eRank collapse has already occurred.
>
> We also track the normalized singular value sum of the Down matrix, $Nesum = \sum_i (\sigma_i/\sigma_{\max})$, which measures spectral uniformity.
>
> | Step | RED ($Nesum$) | LRC ($Nesum$) |
> | --- | ---: | ---: |
> | Init | 1536 | 840 |
> | 20k | 808 | 397 |
> | 180k | 549 | 310 |
> | 200k | 551 | 309 |
>
> **Although RED departs from the theoretical ideal due to nonlinearities, it still stabilizes at a much richer spectrum than LRC.**
>
> > Q3:SFT or RL for math/code reasoning on LRC?
>
> Thank you for this excellent suggestion. To explicitly test recoverability, we perform full-parameter SFT on LRC-1.5B using MetaMathQA-40K. We will conduct  RL-specific recovery experiments in the revised version.
>
> | Model | SFT Data | GSM8K |
> | --- | --- | ---: |
> | LRC-Base | None | 0.21 |
> | LRC-SFT | UltraChat | 0.18 |
> | LRC-SFT | MetaMathQA | 0.33 |
> | RED-Base | None | 0.44 |
>
> **Targeted math SFT mitigates the degradation in LRC, but it remains below RED-Base. Furthermore, the eRank of the MetaMath-SFT model remains substantially collapsed**. This supports our claim that addressing the structural eRank collapse at initialization is more effective.
>
> > Q4:RED on harder benchmarks with CoT?
>
> We agree that evaluating on more challenging reasoning benchmarks is important. We additionally evaluate Math500 and two challenging BBH tasks under CoT-style prompting.
>
> | Model | Math500 | BBH (Bool. Expr.) | BBH (Geom. Shapes) |
> | --- | ---: | ---: | ---: |
> | LRC | 0.02 | 0.63 | 0.22 |
> | RED | 0.08 | 0.68 | 0.30 |
>
> **RED shows consistent improvements over LRC on these harder tasks.** ARC-Challenge CoT is not currently supported natively in the lm-eval setup, but we will expand this coverage in the revision.
>
> >W2: Depth-reduced analysis is shallower.
>
> We sincerely thank the reviewer for highlighting this scope issue. We agree that, in the current paper, our theoretical analysis and proposed remedy are specifically targeted at the width-reduced setting, **as empirical evidence shows width-pruning generally preserves a higher performance ceiling**.
>
> We include the depth-reduced analysis to show that reasoning collapse is a broader phenomenon across the EDistill paradigm, but **depth reduction likely requires different solutions because it disrupts the layer-wise gradual refinement process that multi-step reasoning relies on (Appendix E)**. We hope that we can motivate future work on depth-specific remedies. A recently published paper employs a lookup table approach to bypass the first few layers of LLMs, allowing them to focus exclusively on reasoning.

---

> > ### Author Rebuttal · Reviewer_8tgS · 2026-04-03
> >
> > The rebuttal is reasonable, my concerns are addressed

---

> > > ### Author Response · Authors · 2026-04-03
> > >
> > > Thank you once again for your positive feedback and insightful suggestions. We are very grateful for the time and effort you have put into this.

---

### Decision · Program_Chairs · 2026-04-30

**Decision:**

Accept (regular)

**Comment:**

This paper identifies and investigates "reasoning collapse" in width-reduced Efficient Distillation (EDistill) of LLMs, where compressed models retain general ability but severely degrade on multi-step reasoning tasks. The authors trace this to eRank collapse of randomly initialized projection matrices and propose RED, which uses activation-aware initialization to stabilize singular value dynamics. The theoretical analysis is thorough, with formal proofs connecting random initialization to skewed projection spectra, bounded token indistinguishability, and reasoning degradation (Theorems 4.2–4.4). Empirically, RED achieves strong results: at the ~2B scale, it reaches 0.30 reasoning average vs. 0.21 for the LRC baseline while matching general ability (0.59 vs. 0.57), and RED-1.5B reaches GSM8K 0.27 at only 2.8B training tokens — exceeding LRC's final performance at 10B tokens. The rebuttal further strengthened the paper with direct comparisons against orthogonal projection methods (RED eRank 263.28 vs. 149.97 for orthogonal init) and results on harder benchmarks (Math500: 0.22 vs. 0.02 for LRC).

The theoretical analysis relies on a linear autoencoder proxy that strips away the nonlinearities, residual connections, and normalization present in actual EDistill. While the empirical spectral uniformity tracking and singular value correlation data show the theory is directionally correct, the quantitative gap between the idealized model and real training dynamics remains uncharacterized. Additionally, the current scope is limited to width-reduced EDistill; the paper acknowledges that depth-reduced reasoning collapse likely requires different solutions but does not explore them. I encourage the authors to tighten the theory-practice connection with more quantitative validation and consider extending the framework to depth-reduced settings in future work.